# Fast Trainable Projection for Robust Fine-Tuning

**Junjiao Tian**
Georgia Institute of Technology
`jtian73@gatech.edu`

**Yen-Cheng Liu**
Georgia Institute of Technology
`ycliu@gatech.edu`

**James Seale Smith**
Georgia Institute of Technology
`jamessealesmith@gatech.edu`

**Zsolt Kira**
Georgia Institute of Technology
`zkira@gatech.edu`

## Abstract

Robust fine-tuning aims to achieve competitive in-distribution (ID) performance while maintaining the out-of-distribution (OOD) robustness of a pre-trained model when transferring it to a downstream task. Recently, projected gradient descent has been successfully used in robust fine-tuning by constraining the deviation from the initialization of the fine-tuned model explicitly through projection. However, algorithmically, two limitations prevent this method from being adopted more widely, *scalability* and *efficiency*. In this paper, we propose a new projection-based fine-tuning algorithm, Fast Trainable Projection (FTP) for computationally efficient learning of per-layer projection constraints, resulting in an average 35% speedup on our benchmarks compared to prior works. FTP can be combined with existing optimizers such as AdamW, and be used in a plug-and-play fashion. Finally, we show that FTP is a special instance of hyper-optimizers that tune the hyper-parameters of optimizers in a learnable manner through nested differentiation. Empirically, we show superior robustness on OOD datasets, including domain shifts and natural corruptions, across four different vision tasks with five different pre-trained models. Additionally, we demonstrate that FTP is broadly applicable and beneficial to other learning scenarios such as low-label and continual learning settings thanks to its easy adaptability. The code will be available at `https://github.com/GT-RIPL/FTP.git`.

## 1 Introduction

With new progress being made in pre-training of foundation models every year, such as self-supervised [1, 2, 3] or language-supervised training [4], their potential has gone far beyond merely speeding up convergence [5]. They have demonstrated superior transferability to other tasks, reducing the need for data and improving robustness and generalization capabilities [6, 7, 8]. The problem of how to fine-tune (transfer) a foundation model such that we maintain its robustness and generalization capabilities acquired during pre-training on large datasets has therefore become an essential research topic. This problem is hard because the conventional machine learning paradigm of validating on held-out training data does not impose any constraints on robustness and generalization w.r.t. the foundation models. For example, fine-tuning with a slightly large learning rate can easily destroy capabilities that reside in the foundation models [8], while performing well on the target task.

To maintain the robustness and generalization capability of the pre-trained model when fine-tuning, recent projection-based methods explicitly constrain the distance between the fine-tuned and the pre-trained models through projection. For example, MARS-SP [9] specifies a distance constraint shared by all layers in a neural network. However, it is practically intractable to tune a constraint for each layer (poor *scalability*). TPGM [10] proposes to *automatically* learn different constraints

37th Conference on Neural Information Processing Systems (NeurIPS 2023).

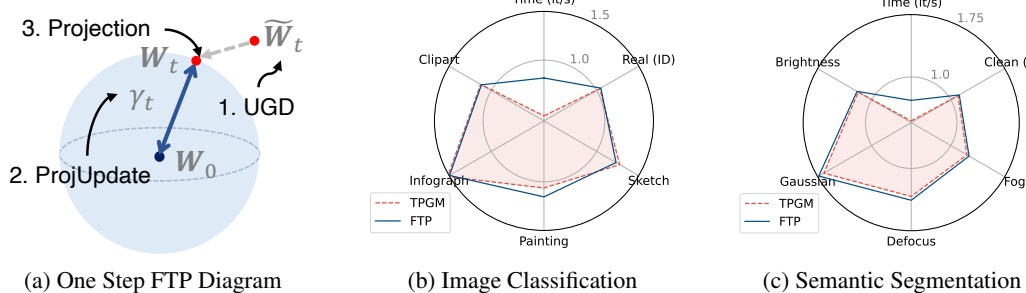

|  (a) One Step FTP Diagram | (b) Image Classification | (c) Semantic Segmentation |

Figure 1: (a): FTP updates the model using (unconstrained) gradient descent (**UGD**) to calculate $\tilde{\mathbf{W}}_t$, then updates the projection constraint $\gamma_t$ (**ProjUpdate**), and finally projects $\tilde{\mathbf{W}}_t$ to $\mathbf{W}_t$ (**Projection**), all in a single forward pass. (b),(c): Visualizations of in-distribution (Real/Clean, labeled as ID), out-of-distribution (Sketch/Fog, etc.) accuracy and computation time (iterations/sec) as a percentage of vanilla fine-tuning (FT) for classification on DomainNet (Tab. 2) and semantic segmentation on PASCAL-Context (Tab. 4) respectively. FTP improves the OOD robustness of FT and is much more computationally efficient than prior work TPGM.

for each layer, solving the issue of scalability in MARS-SP, however, with increased computational overhead (poor *efficiency*). These limitations prevent the method from being adopted more widely.

To achieve scalability and efficiency simultaneously, we propose Fast Trainable Projection (FTP), for learning both the projection constraints and the main model in a single forward pass (Fig. 1a), significantly reducing computation overhead in prior works while achieving competitive performance. Specifically, FTP removes the algorithmic redundancy of extra training procedures required in TPGM [10], which requires sampling a separate batch of data and running a nested training loop. FTP achieves this by 1) utilizing different batches of training data sampled at consecutive steps and 2) re-using gradients calculated for the main model update (Sec. 3.2). This leads to a $35\%$ speedup with comparable performance in fine-tuning (Fig. 1b, 1c). The efficiency improvement and easy adaptability as a drop-in replacement with existing optimizers are essential to making projection-based methods applicable to more fine-tuning problems. For example, we implement SGDP, an SGD variant with built-in FTP. SGDP can be used as a drop-in replacement for SGD (details in Appendix 8.7) as:

```
optimiser = SGDP(param_group,**optimizer_params) #See Appendix 8.7.
```

To demonstrate this, we test FTP on four different vision tasks, image classification, semantic segmentation, human parts segmentation, and surface normal estimation. FTP shows superior OOD performance under domain shift or natural corruptions on all benchmarks. Moreover, we apply FTP to a continual learning (CL) benchmark and achieve state of the art performance when combined with a simple CL technique.

Finally, we show that FTP is a special instance of hyper-optimizers [11, 12, 13, 14, 15, 16, 17, 18] that aims to reduce the manual tuning of optimization hyper-parameters such as learning rate by learning them automatically through automatic differentiation and nested optimization. Theoretically, to understand why FTP and other projection methods can maintain the robustness of the pre-trained models, we propose to establish a theoretical connection between robustness and projection through the lens of Liptschitz continuity, a widely adopted measure of robustness [19, 20, 21]. In summary, our contributions are:

- We present a new fine-tuning algorithm, Fast Trainable Projection, to efficiently learn the projection constraints and fine-tune the model simultaneously, bringing significantly improved computation efficiency w.r.t. prior works [10] in Sec. 3.2.

- We show that FTP is a special instance of hyper-optimizers that aims to reduce manual tuning of hyper-parameters through nested optimization in Sec. 3.3.

- We discuss a dual perspective of the fine-tuning robustness in the feature space and the weight space of a model to mathematically understand why projection can maintain the robustness of the pre-trained models in Sec. 3.4.

- We show superior robustness on OOD datasets on four vision tasks with five pre-trained models and SOTA performance on a continual learning benchmark, all with a 35% speedup in Sec. 4.

## 2   Related Works

We summarize related works in (general) robust fine-tuning into three categories: when, where, and how much to fine-tune, depending on their underlying strategy. Moreover, we discuss recent advances in fine-tuning language-image pre-trained models, which have inspired specialized fine-tuning strategies. **When to fine-tune:** LP-FT [7] discovers that fine-tuning the entire network can distort features in the pre-trained models and proposes to first only fine-tune the last linear layer followed by training the entire network with a small learning rate. We will include LP-FT in our experiments. **Where to fine-tune:** Instead of fine-tuning the entire network, some methods investigate the choice of weights to fine-tune. SpotTune [22] learns where to fine-tune through an additional policy network. However, SpotTune needs to retain the policy network, the pre-trained model, and the fine-tuned model in memory for inference, adding significant computation at inference time. Recently, SurgicalFT [23] proposes to use the GradientNorm heuristic, the ratio of the gradient norm to the parameter norm, to determine which layer to fine-tune. Parameter-efficient fine-tuning methods are another example of this category. While they aim to minimize the parameters tuned, they have been shown to improve OOD generalization performance as well in NLP applications [24, 25, 26, 27, 28, 29]. We specifically compare to two recent parameter-efficient methods that only tune the bias terms: Bitfit [30] for Transformers [31] and Partial Fusion [32] for ResNets [33]. **How much to fine-tune:** Our work belongs to this category where the entire neural network is fine-tuned simultaneously. Specifically, we can split works into two sub-categories: regularization and projections. **Regularization:** DELTA [34] proposes to regularize the output (feature maps) of a neural network to that of its pre-trained model. This requires two separate passes through the pre-trained model and the fine-tuned model increasing both memory and computation overhead. L2-SP [35] instead regularizes the L2 distance between the fine-tuned model and the pre-trained model, serving as a strong baseline. **Projection:** Utilizing projection to enforce a close distance to the pre-trained model has been studied in prior works: MARS-SP [9] and TPGM [10]. We dedicate a section to revisit them later in the method section (Sec. 3.1). **Language-Image Pre-trained Models.** Several recent works have proposed special fine-tuning strategies for language-image pre-trained models with zero-shot capability. WISE-FT [8] achieves SOTA performance by linearly interpolating a fine-tuned model and its initialization *at the end* of fine-tuning. However, it only applies to a subset of pre-trained models with linear connectivity such as CLIP [4]. FT-Like-Pretrain [36] proposes to use a contrastive fine-tuning strategy, the same strategy used in pre-training for those models, instead of the conventional cross-entropy loss for many vision tasks. The method has demonstrated superior results when combined with WISE-FT, where WISE-FT contributes the most to the improvement. Similarly, we will also combine our method with WISE-FT to show improved OOD performance using CLIP.

## 3   Method

### 3.1   Review: Enforcing Projection and Learning Constraints

In this work, we focus on fine-tuning a pre-trained model, where $\mathbf{W}_0 \in \mathbb{R}^{n \times m}$ is the weights of a linear layer in the pre-trained model, to a downstream task. We denote $\mathbf{W}_t$ as the fine-tuned model at training iteration $t$ and $\mathbf{w}^i$ as the $i$th row of a matrix $\mathbf{W} \in \mathbb{R}^{n \times m}$. Several prior works [9, 10] have attempted to use projection to improve fine-tuning robustness. The most vanilla formulation in MARS-SP [9] has two steps: *unconstrained gradient descent* and *projection*.

**Unconstrained Gradient Descent** (Abbrev. UGD), the projection-based methods first compute the updated model weights $\tilde{\mathbf{W}}_t$ *without* projection. For example, at iteration $t$, given a batch of training data $\mathcal{D}_t^{tr}$, we first obtain $\tilde{\mathbf{W}}_t$ as the following,

$$\mathbf{g}_t \leftarrow \nabla \mathcal{L}_{\mathcal{D}_t^{tr}}(\mathbf{W}_{t-1}) \in \mathbb{R}^{n \times m}, \quad \tilde{\mathbf{W}}_t \leftarrow \mathbf{Opt}(\mathbf{W}_{t-1}, \mathbf{g}_t). \tag{1}$$

where $\mathbf{g}_t$ is the derivative of the loss function $\mathcal{L}_{\mathcal{D}_t^{tr}}(\mathbf{W}_{t-1})$ calculated on $\mathcal{D}_t^{tr}$ w.r.t. $\mathbf{W}_{t-1}$ and $\mathbf{Opt}(\cdot)$ is an existing optimization algorithm, such as SGD, AdamW [37].

**Projection.** MARS-SP [21] projects the updated model $\tilde{\mathbf{W}}_t$ towards its initialization $\mathbf{W}_0$ with a pre-defined projection constraint $\gamma$ for all layers using the MARS matrix norm (see Appendix . 8.2) as shown below in Eq. 2.

$$\mathbf{W}_t = \Pi(\tilde{\mathbf{W}}_t, \mathbf{W}_0, \gamma_t) = \begin{bmatrix} \mathbf{w}_t^{i\mathsf{T}} \\ \vdots \\ \mathbf{w}_t^{i\mathsf{T}} \end{bmatrix} = \begin{bmatrix} \left( \frac{\gamma}{\|\tilde{\mathbf{w}}_t^i - \mathbf{w}_0^i\|_1}(\tilde{\mathbf{w}}_t^i - \mathbf{w}_0^i) + \mathbf{w}_0^i \right)^{\mathsf{T}} \\ \vdots \\ \left( \frac{\gamma}{\|\tilde{\mathbf{w}}_t^n - \mathbf{w}_0^n\|_1}(\tilde{\mathbf{w}}_t^n - \mathbf{w}_0^n) + \mathbf{w}_0^n \right)^{\mathsf{T}} \end{bmatrix} \tag{2}$$

However, MARS-SP has poor scalability because it is practically intractable to hand-tune different constraints for each layer, which results in sub-optimal performance as reported by TPGM [10]. Instead of a pre-defined $\gamma$ for all layers, TPGM proposes to learn a different constraint $\gamma_t$ for each layer[1]. and updates them iteratively during training. This enables TPGM to customize a different regularization strength for each layer and to have superior performance for both ID and OOD data.

**ProjUpdate.** Given as input the frozen unconstrained model $\tilde{\mathbf{W}}_t$ from UGD (Eq. 1), TPGM adds an intermediate *ProjUpdate* function before *projection*, which samples a separate set of data from the validation dataset $\mathcal{D}_t^{val}$ and uses a standalone training loop to update the projection constraints $\gamma_t$ while keeping the model $\tilde{\mathbf{W}}_t$ frozen. Specifically, *ProjUpdate* creates a temporary projected model $\mathbf{W}_p$ by projecting $\tilde{\mathbf{W}}_t$ towards $\mathbf{W}_0$ based on the previous constraint $\gamma_{t-1}$ using Eq. 2, i.e., $\mathbf{W}_p = \Pi(\;\tilde{\mathbf{W}}_t\;, \mathbf{W}_0, \gamma_{t-1})$. Therefore, $\mathbf{W}_p(\gamma_{t-1})^2$ can be viewed as a function of $\gamma_{t-1}$. Then FTP calculates the gradient $\nabla\gamma_t$ by taking a derivative of the loss function $\mathcal{L}_{\mathcal{D}_t^{val}}(\mathbf{W}_p(\gamma_{t-1}))$ w.r.t. $\gamma_{t-1}$:

$$\nabla\gamma_t \leftarrow \nabla\mathcal{L}_{\mathcal{D}_t^{val}}(\mathbf{W}_p(\gamma_{t-1})), \quad \gamma_t = \mathbf{Opt}(\gamma_{t-1}, \nabla\gamma_t) \tag{3}$$
$$\text{where} \quad \mathbf{W}_p = [\mathbf{w}_p^i, \ldots, \mathbf{w}_p^n]^{\mathsf{T}} \quad \text{and} \quad \mathbf{w}_p^i = \frac{\gamma_{t-1}}{\|\tilde{\mathbf{w}}_t^i - \mathbf{w}_0^i\|_1}(\tilde{\mathbf{w}}_t^i - \mathbf{w}_0^i) + \mathbf{w}_0^i.$$

With the calculated gradient, TPGM uses an existing optimizer $\mathbf{Opt}(\cdot)$ to update $\gamma_t$. This procedure, sampling $\mathcal{D}_t^{val}$ and calculating the derivative $\nabla\mathcal{L}_{\mathcal{D}_t^{val}}(\mathbf{W}_p(\gamma_{t-1}))$, is the key to learning projection constraints because the unconstrained model $\tilde{\mathbf{W}}_t$ (highlighted above and calculated in Eq. 1), was updated on the training data $\mathcal{D}_t^{tr}$ and $\gamma_t$ is now updated on separate data $\mathcal{D}_t^{val}$. The discrepancy between $\mathcal{D}_t^{tr}$ and $\mathcal{D}_t^{val}$ allows TPGM to find a better projected model $\mathbf{W}_p$ (projected between $\tilde{\mathbf{W}}_t$ and $\mathbf{W}_0$) by updating $\gamma_t$, which balances between fitting the training data $\mathcal{D}_t^{tr}$ and generalizing to $\mathcal{D}^{val}$. Finally, with an updated $\gamma_t$, TPGM *again* projects $\tilde{\mathbf{W}}_t$ towards $\mathbf{W}_0$ to obtain the final model $\mathbf{W}_t$ using Eq. 2, replacing the pre-defined $\gamma$ with a learned $\gamma_t$. A flow chart of TPGM is in Fig. 2.

The algorithm demonstrated the capability to automatically learn different constraints for each layer, solving the scalability issue in MARS-SP. However, TPGM introduces extra computation in the additional training loop. In the next section, we propose a scalable and efficient projection algorithm that learns the projection constraints for each layer without separate validation data and loops.

### 3.2 FTP: Fast Trainable Projection

To inherit the scalability of TPGM while reducing the computational overhead, we propose Fast Trainable Projection (FTP) (Algorithm 1). Similar to TPGM, the algorithm has three components: *UGD*, *ProjUpdate*, *Projection*. The *ProjUpdate* component is the major contributor to efficient computation. It builds on a key insight: Instead of sampling separate data $\mathcal{D}_t^{val}$ each time, we use two training data batches sampled independently at consecutive steps, e.g., $\mathcal{D}_{t-1}^{tr}$ and $\mathcal{D}_t^{tr}$. Specifically, we use $\mathcal{D}_t^{tr}$ to update $\gamma_t$ instead of $\mathcal{D}_t^{val}$. As a result, the optimization of $\gamma_t$ re-uses most of the computation used for the optimization of the main model.

**ProjUpdate.** Specifically, instead of taking a derivative of $\mathcal{L}_{\mathcal{D}_t^{val}}(\mathbf{W}_p)$ w.r.t. $\gamma_{t-1}$ as in TPGM, FTP calculates the gradient of $\gamma_{t-1}$ by the derivative of the loss function on the current training data $\mathcal{L}_{\mathcal{D}_t^{tr}}(\mathbf{W}_{t-1})$ w.r.t. $\gamma_{t-1}$. Note that $\mathbf{W}_{t-1} = \Pi(\;\tilde{\mathbf{W}}_{t-1}\;, \mathbf{W}_0, \gamma_{t-1})$ is also a function of the

---

[1]We omit the index for different layers to avoid notation clutter and the subscript $t$ indicates training iterations.
[2]We use this functional form $\mathbf{W}_p(\gamma_{t-1})$ to highlight the dependency on $\gamma_{t-1}$.

---

**Algorithm 1** FTP: **F**ast **T**rainable **P**rojection.

---

**Require:** $\mathbf{W}_0$ the pre-trained model
**Require:** $\kappa$ positive gradient annealing rate
**Require:** $\mu \leftarrow 1e-2, \beta_1, \beta_2 \leftarrow (0.9, 0.999)$ fixed parameters for AdamUpdate

    **for** $t = 1...T$ **do**

$$\begin{cases} \mathbf{g}_t \leftarrow \nabla \mathcal{L}_{\mathcal{D}^{tr}}(\mathbf{W}_{t-1}) \\ \tilde{\mathbf{W}}_t \leftarrow \mathbf{Opt}(\mathbf{W}_{t-1}, \mathbf{g}_t) \end{cases} \qquad \triangleright \text{ Unconstrained Gradient Descent (Eq. 1)}$$

        **if** $t = 1$ **then**
            $\gamma_t = 1e-8$                                                            $\triangleright$ Initialize $\gamma$
        **else**

$$\begin{cases} \nabla\gamma_t \leftarrow \sum_i \left( \mathbf{g}_t^{i,\intercal}(\tilde{\mathbf{w}}_{t-1}^i - \mathbf{w}_0^i) \frac{1}{\|\tilde{\mathbf{w}}_{t-1}^i - \mathbf{w}_0^i\|_1} \right) \\ \mathbf{if} \ \nabla\gamma_t > 0 : \nabla\gamma_t = \kappa\nabla\gamma_t \\ \gamma_t \leftarrow \mathbf{AdamUpdate}(\gamma_{t-1}, \nabla\gamma_t, t) \end{cases} \qquad \triangleright \text{ ProjUpdate (Eq. 4,Eq. 5, Alg. 2)}$$

       $\mathbf{W}_t = \Pi(\tilde{\mathbf{W}}_t, \mathbf{W}_0, \gamma_t)$                                          $\triangleright$ Projection (Eq. 2)

---

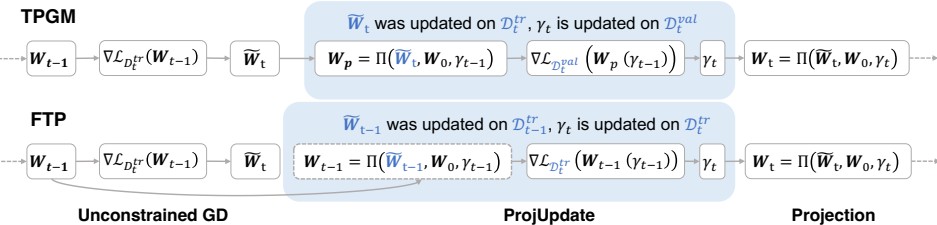

Figure 2: Computation Flow Chart of TPGM (top) and FTP (bottom) at iteration $t$. The main difference between TPGM and FTP is in the **PorjUpdate** step. FTP uses the previous model $\mathbf{W}_{t-1}$ and cached gradients from $\mathcal{L}_{\mathcal{D}_{tr}^t}(\mathbf{W}_{t-1})$ to update the projection constraints $\gamma_t$.

constraint $\gamma_{t-1}$ as a result of projection from the previous step. Hence, by virtue of the chain rule, the gradient of $\mathcal{L}_{\mathcal{D}^{tr}}(\mathbf{W}_{t-1}(\gamma_{t-1}))$ w.r.t. $\gamma_{t-1}$ is,

$$\nabla\gamma_t = \sum_{i=1}^{n} \nabla \underbrace{\mathcal{L}_{\mathcal{D}_t^{tr}}(\mathbf{w}_{t-1}^i(\gamma_{t-1}))^{\intercal}}_{\mathbf{g}_t^i} \frac{\partial \mathbf{w}_{t-1}^i}{\partial \gamma_{t-1}} = \sum_{i=1}^{n} \mathbf{g}_t^{i,\intercal}(\tilde{\mathbf{w}}_{t-1}^i - \mathbf{w}_0^i) \frac{1}{\|\tilde{\mathbf{w}}_{t-1}^i - \mathbf{w}_0^i\|_1} \qquad (4)$$

where the summation loops over each row in the matrix $\mathbf{W}_{t-1}$ because the same constraint $\gamma_{t-1}$ is enforced for all rows (see MARS norm in Appendix Eq. 15 and Eq. 2 ) so the final gradient is the summation of all gradients for each row. Similar to TPGM, because the starting point of projection $\tilde{\mathbf{W}}_{t-1}$ (highlighted above) was updated using the previous training batch $\mathcal{D}_{t-1}^{tr}$ and the gradient $\nabla\gamma_t$ is calculated using the current batch $\mathcal{D}_t^{tr}$, the discrepancy between $\mathcal{D}_{t-1}^{tr}$ and $\mathcal{D}_t^{tr}$ enables FTP to learn meaningful projection constraints. Crucially, we proposed a novel formulation that allows for **re-using** the gradient $\mathbf{g}_t$ used for calculating the unconstrained model $\tilde{\mathbf{W}}_t$ in the UGD step (Eq. 1).

**Gradient Annealing.** Prior work [10] noticed that learning projection constraints for each layer can suffer from underfitting because the learned constraints can be too conservative, and used an additional regularization to help reduce this negative effect. For FTP, we introduce a simple technique that uses a single gradient annealing factor for all layers, $\kappa \in [0, 1]$ to modulate the magnitude of the *positive* gradient $\nabla\gamma_t > 0$, which contributes to the shrinkage of the constraints. When $\nabla\gamma_t > 0$,

$$\nabla\gamma_t = \kappa\nabla\gamma_t. \qquad (5)$$

For example, when $\kappa = 0$, the projection constraint $\gamma$ will not receive any positive gradient and is therefore non-decreasing. With the annealed gradients $\nabla\gamma_t$, we update the constraint using the Adam update rule [38] because Adam is suitable for non-stationary optimization, where the optimal values change over time. Please see Appendix 8.3 for a detailed algorithmic description of **AdamUpdate** and additional discussion on how FTP saves computation.

Finally, after obtaining the updated $\gamma_t$ from **AdamUpdate**, FTP applies the learned constraints to project the current unconstrained model $\tilde{\mathbf{W}}_t$ towards the pre-trained model $\mathbf{W}_0$ using Eq. 2 with a different constraint for each layer. The complete algorithm is summarized in Alg. 1. For a quick comparison with TPGM, we provide a side-by-side computation flow chart of FTP in Fig. 2.

**Implicit Assumption.** The algorithmic difference between TPGM and FTP makes an implicit assumption. Specifically, after obtaining the updated constraints $\gamma_t$ after **AdamUpdate**, if the algorithm were to follow TPGM, the next step would be applying the updated constraints to *re-calculate* the previous model $\mathbf{W}_{t-1}$ since $\gamma_t$ is updated based on $\mathbf{W}_{t-1}$ (Eq. 4). However, instead of rolling back, FTP applies the updated constraints directly to the current unconstrained model $\tilde{\mathbf{W}}_t$ to calculate $\mathbf{W}_t$. This step assumes *smoothness* in the update of $\gamma_t$, i.e., the $\gamma_t$ does not change drastically in consecutive steps. The assumption is valid since $\gamma_t$ is updated by *AdampUpdate* (Alg. 2 in Appendix 8.3) which uses a moving average update with a momentum of $0.9$. So the change of $\gamma_t$ is very smooth because of the high discount factor of $0.9$. Importantly, it enables us to **re-use** the same gradient $\mathbf{g}_t$ available for computing the current unconstrained model $\tilde{\mathbf{W}}_t$ to update $\gamma_t$. This is the key to saving computation because the separate training loop as a result of "rolling back" is the main computation bottleneck in TPGM.

### 3.3 FTP as a Hyper-Optimizer for Fine-Tuning

The FTP algorithm in Alg. 1 bears motivational and algorithmic similarity to a recent resurrection of *hyper-optimizers* [11, 12, 13, 14, 15, 16, 17, 18]. Specifically, hyper-optimizers aim to learn the hyper-parameters such as the learning rate in an optimizer by treating them as learnable parameters through nested differentiation and optimization because manual tuning of those hyper-parameters can be time-consuming and can lead to sub-optimal performance. FTP stems from the same motivation as the manual specification of projection constraints can be computationally infeasible [10].

To understand the algorithmic similarity better, let's use SGD as an example. Suppose at iteration $t-1$, we have updated the model parameters $\mathbf{W}_{t-2}$ through SGD with a learning rate $\alpha_{t-1}$.

$$\mathbf{W}_{t-1} = \mathbf{W}_{t-2} - \alpha_{t-1}\nabla\mathcal{L}(\mathbf{W}_{t-2}) \tag{6}$$

At the current step $t$, hyper-optimizers first calculate the gradient w.r.t to the learning rate $\alpha_{t-1}$ and update it using another SGD optimizer with a new learning rate parameter $\kappa$.

$$\alpha_t = \alpha_{t-1} - \kappa\frac{\partial\mathcal{L}(\mathbf{W}_{t-1})}{\partial\alpha} = \alpha_{t-1} + \kappa\nabla\mathcal{L}(\mathbf{W}_{t-1})^T\nabla\mathcal{L}(\mathbf{W}_{t-2}) \tag{7}$$

Finally, using the updated $\alpha_t$, hyper-optimizers update the main model parameters.

$$\mathbf{W}_t = \mathbf{W}_{t-1} - \alpha_t\nabla\mathcal{L}(\mathbf{W}_{t-1}) \tag{8}$$

It's not hard to spot the algorithmic similarity between the FTP algorithm and hyper-optimizers. Both algorithms first update the hyper-parameters (projection constraints $\gamma_t$ in Eq. 4 vs. the learning rate $\alpha_t$ in Eq. 7) using the cached information from the previous iteration and the gradient from the current iteration (known as hyper-gradients). Then, they apply the updated hyper-parameters to calculate the current model. Finally, hyper-optimizers make the same assumption of smoothness in the update of the hyper-parameters such that the update of the hyper-parameters and the model parameters can be performed consecutively in a single forward pass. In this regard, FTP can be seen as a special instance of hyper-optimizer for fine-tuning.

### 3.4 Dual Perspective: Fine-tuning Robustness in Feature Space and Weight Space

It is not immediately clear why FTP's and other methods' projections in the weight space maintain the robustness of the pre-trained model in the feature space besides the intuition that the closer to the pre-train model the more likely the fine-tuned model will behave like it. To fully understand this, we study the mathematical connection between projection and robustness. Let $\mathbf{x} \in \mathbb{R}^m$ denote an input vector and $h(\mathbf{x}) : \mathbb{R}^m \to \mathbb{R}^n$ a function mapping it to a feature space. Given two input vectors $\mathbf{x}, \mathbf{x}' \in \mathbb{R}^m$, we denote the distance between them in the original space by their vector norms $\|\mathbf{x} - \mathbf{x}'\|_{\mathrm{x}}$ and in the feature space by $\|h(\mathbf{x}) - h(\mathbf{x}')\|_{\mathrm{h}}$. Let $h_f(\cdot)$ and $h_0(\cdot)$ denote a fine-tuned model and its pre-trained initialization, and $\Delta h(\cdot) \equiv h_f(\cdot) - h_0(\cdot)$ denotes the *difference function*.

To capture the robustness of a fine-tuned model, we apply the notion of Lipschitz continuity on the *difference function* because its Lipschitz constant captures the maximum rate of change of differences between the fine-tuned model and the pre-trained model in the feature space. Formally,

$$\|\Delta h(\mathbf{x}) - \Delta h(\mathbf{x}')\|_{\mathrm{h}} \leq L_d \|\mathbf{x} - \mathbf{x}'\|_{\mathrm{x}} \quad \forall (\mathbf{x}, \mathbf{x}') \in \mathbb{R}^m. \tag{9}$$

where $L_d \geq 0$ is the Lipschitz constant of the difference function $\Delta h(\mathbf{x})$. If the inequality is satisfied, in this paper, we call $h_f(\cdot)$ $L_d$-Lipschitz-robust w.r.t. the pre-trained initialization $h_0(\cdot)$.

The definition has a natural intuition stemming from Lipschitz continuity, a measure of robustness [19, 20, 21]. A Lipschitz function is limited by how fast it can change, governed by the Lipschitz constant. Traditionally, a small Lipschitz constant is associated with better robustness, because a small constant means less sensitivity to changes in the input. We provide the following lemma (proof in Appendix 8.1) to illustrate the connection between the *difference function* and the robustness of the fine-tuned model.

**Lemma 1.** *If a fine-tuned model $h_f(\cdot)$ is $L_d$-Lipschitz-robust with respect to its $L_0$-Lipschitz pre-trained initialization $h_0(\cdot)$, i.e., $\forall (\mathbf{x}, \mathbf{x}') \in \mathbb{R}^m$,*

$$\|\Delta h(\mathbf{x}) - \Delta h(\mathbf{x}')\|_{\mathrm{h}} \leq L_d \|\mathbf{x} - \mathbf{x}'\|_{\mathrm{x}} \quad and \quad \|h(\mathbf{x})_0 - h(\mathbf{x}')_0\|_{\mathrm{h}} \leq L_0 \|\mathbf{x} - \mathbf{x}'\|_{\mathrm{x}}$$

*then, $h_f(\cdot)$ is $(L_d + L_0)$-Lipschitz, i.e.,*

$$\|h_f(\mathbf{x}) - h_f(\mathbf{x}')\|_{\mathrm{h}} \leq (L_d + L_0)\|\mathbf{x} - \mathbf{x}'\|_{\mathrm{x}} \quad \forall (\mathbf{x}, \mathbf{x}') \in \mathbb{R}^m.$$

**Feature Space.** From Lemma 1, we can see that minimizing $L_d$ can improve the robustness of the fine-tuned model, defined by its Lipschitz constant $(L_d + L_0)$, which equals $L_0$ when $L_d = 0$. Therefore, the fine-tuned model can achieve a similar level of robustness as the pre-trained model if $L_d$ is minimized. Colloquially, given two inputs $(\mathbf{x}, \mathbf{x}')$, where $\mathbf{x}'$ is a perturbed version of $\mathbf{x}$, $h_f(\cdot)$ will be just as sensitive/robust to the perturbation as $h_0(\cdot)$ is if $L_d$ is small.

**Weight Space.** The definition of fine-tuning robustness (Eq. 9) not only leads to an interpretation of robustness in the feature space (Lemma 1) but also conveniently a *projection* operation in the weight space. Specifically, we investigate a single linear layer in a neural network and show that enforcing the inequality in Eq. 9 leads to a projection operation by virtue of linear operators and matrix norms. We illustrate this in the following lemma with a full discussion and proof in Appendix 8.2.

**Lemma 2.** *Assuming linear models $h(\mathbf{x}) = \mathbf{W}\mathbf{x} + \mathbf{b}, \mathbf{W} \in \mathbb{R}^{n \times m}, \mathbf{b} \in \mathbb{R}^n$, and both the input space vector norm $\|\cdot\|_{\mathrm{x}}$ and the feature space vector norm $\|\cdot\|_{\mathrm{h}}$ are defined by $l_\infty$ norm. $\mathbf{w}_p^i$ satisfies the inequality in Eq. 9 if*

$$\mathbf{w}_p^i = \min\left(1, \frac{L_d}{\widetilde{\|\mathbf{w}_f^i - \mathbf{w}_0^i\|_1}}\right)(\mathbf{w}_f^i - \mathbf{w}_0^i) + \mathbf{w}_0^i, \quad \forall i \in \{1, ..., n\}. \tag{10}$$

*where $\mathbf{w}^i$ denotes the $i$-th row of the matrix $\mathbf{W}$ and $\mathbf{w}_p^i$ is the new projected fine-tuned model.*

This is an equation of *projection* between $\mathbf{W}_f$ and $\mathbf{W}_0$ defined by the MARS norm in the weight space for a single linear layer and is the projection operation used in FTP and prior works [9, 10] (Eq. 2). It indicates that we can choose an arbitrarily small $L_d$ and enforce it through Eq. 10, potentially trading off fitting the downstream task and preserving robustness. In summary, this section demonstrates the connection between robustness and projection. Specifically, we have shown that to achieve good fine-tuning robustness, we can enforce a small Lipschitz constant $L_d$ on the difference function $\Delta h(\mathbf{x})$ **in the feature space** (Lemma 1), which can be physically enforced through the projection of the fine-tuned model towards the pre-trained model **in the weight space** (Lemma. 2).

## 4 Experiments

**Overview.** To validate the effectiveness of FTP in fine-tuning pre-trained models, we benchmark FTP on both image classification (Sec. 4.1) and dense vision tasks (Sec. 4.2) with different network architectures and pre-trained models. For each benchmark, we report both in-distribution (ID) performance as well as out-of-distribution (OOD) performance. We show that FTP not only achieves competitive ID performance and superior OOD performance but is also much more computationally efficient than prior works. We further test FTP's regularization capability on a continual learning benchmark and show state of art performance against recent SOTA methods (Sec. 4.3).

Table 1: DomainNet Results using MOCO-V3 pre-trained ResNet50 with 100% Real Data. FTP achieves the best OOD performance and is much faster than prior work TPGM [10] by **36**%.

| | ID | OOD | | | | Statistics | | | |
|---|---|---|---|---|---|---|---|---|---|
| | Real | Sketch | Painting | Infograph | Clipart | OOD Avg. | ID Δ (%) | OOD Δ (%) | Time (s/it)↓ |
| Vanilla FT | 81.99 (0.03) | 31.52 (0.33) | 42.89 (0.53) | 18.51 (0.28) | 44.98 (0.24) | 34.47 | 0.00 | 0.00 | 0.35 |
| Linear Prob. | 73.01 (0.03) | 24.10 (0.23) | 39.56 (0.15) | 12.27 (0.02) | 30.38 (0.08) | 26.58 | -10.96 | -22.90 | 0.10 |
| Partial Fusion [32] | 78.27 (0.03) | 27.72 (0.07) | 39.74 (0.12) | 15.56 (0.08) | 38.18 (0.12) | 30.30 | -4.55 | -12.11 | 0.21 |
| L2-SP [35] | 81.51 (0.02) | 34.91 (0.22) | 45.76 (0.16) | 18.97 (0.11) | 45.29 (0.18) | 36.23 | -0.59 | 5.09 | 0.46 |
| MARS-SP [9] | 81.89 (0.01) | 34.44 (2.54) | 45.05 (1.91) | 19.97 (1.48) | 46.36 (1.29) | 36.45 | -0.13 | 5.74 | 0.43 |
| LP-FT [7] | **82.92** (0.01) | 34.50 (0.22) | 45.42 (0.31) | 20.12 (0.43) | **47.11** (0.27) | 36.79 | 1.13 | 6.72 | - |
| TPGM [10] | 82.66 (0.13) | 35.35 (0.33) | 46.20 (0.20) | 20.13 (0.12) | 45.75 (0.12) | 36.86 | 0.82 | 6.91 | 0.80 |
| FTP | 82.17 (0.02) | **36.26** (0.06) | **46.58** (0.10) | **20.67** (0.03) | 46.97 (0.06) | **37.62** | 0.22 | **9.13** | 0.51 |

Table 2: DomainNet Results using CLIP pre-trained ResNet50 with 100% Real Data. FTP achieves competitive OOD performance and is much faster than prior work TPGM [10] by **36**%.

| | ID | OOD | | | | Statistics | | | |
|---|---|---|---|---|---|---|---|---|---|
| | Real | Sketch | Painting | Infograph | Clipart | OOD Avg. | ID Δ (%) | OOD Δ (%) | Time (s/it)↓ |
| Vanilla FT | 80.93 (0.08) | 31.81 (0.06) | 41.02 (0.10) | 20.29 (0.08) | 43.59 (0.15) | 34.18 | 0.00 | 0.00 | 0.58 |
| Linear Prob. | 52.56 (0.09) | 20.05 (0.21) | 24.92 (2.49) | 19.18 (0.46) | 21.15 (0.18) | 21.33 | -35.05 | -37.60 | 0.14 |
| Partial Fusion [32] | 78.27 (0.11) | 36.77 (0.32) | 42.13 (0.35) | 24.71 (0.18) | 43.31 (0.53) | 36.73 | -3.29 | 7.46 | 0.33 |
| L2-SP [35] | 82.07 (0.09) | 36.67 (0.11) | 45.62 (0.35) | 22.97 (0.42) | 47.78 (0.30) | 38.26 | 1.40 | 11.94 | 0.62 |
| MARS-SP [9] | 77.19 (0.63) | 25.33 (1.07) | 33.43 (2.06) | 14.81 (0.43) | 39.20 (0.74) | 28.19 | -4.62 | -17.53 | 0.61 |
| LP-FT [7] | 80.82 (0.95) | 34.85 (1.93) | 44.03 (0.05) | 22.23 (2.01) | 46.13 (2.34) | 36.81 | -0.14 | 7.69 | - |
| TPGM [10] | 83.64 (0.01) | **38.78** (0.42) | 43.11 (0.25) | **28.70** (0.31) | **48.01** (0.25) | 39.65 | 3.34 | 16.01 | 1.07 |
| FTP | **84.22** (0.11) | 37.66 (0.45) | **46.11** (0.29) | 28.33 (0.33) | 47.67 (0.18) | **39.94** | **4.05** | **16.87** | 0.68 |

## 4.1 Image Classification Experiments

### 4.1.1 DomainNet

For the DomainNet experiment (image classification), which consists of five domains, Real, Sketch, Painting, Infographics, and Clipart, we follow the setup of the prior work [10] and use its released code to train FTP. Specifically, we use two pre-trained models, an ImageNet pre-trained MOCO-V3 ResNet50 [3] and a CLIP pre-trained ResNet50 [4]. For FTP, we only tuned the learning rate while keeping the other hyper-parameters fixed as in the prior work. We use the Real domain as the ID training dataset and the rest as OOD testing datasets. Please refer to Appendix 8.4 for more details.

**FTP achieves the best OOD accuracy and is much more efficient.** In Tab. 1 and Tab. 2, we show results training on 100% DomainNet-Real data using CLIP and MOCO-V3 pre-trained initialization respectively. Compared to the previous SOTA methods TPGM [10], FTP achieves competitive ID accuracy and better OOD generalization performance. *More importantly, in addition to favorable results, FTP is 36% faster on average on both benchmarks compared to TPGM*. Following TPGM [10], we also report results training only on 10% DomainNet-Real data in Appendix Tab. 6.

### 4.1.2 ImageNet

Recently, zero-shot language-vision pre-trained models such as CLIP [4] have demonstrated strong generalization capability to other tasks. Notably, WISE [8] showed that linear interpolation between a fine-tuned model and its initialization achieves significant improvement in OOD generalization.

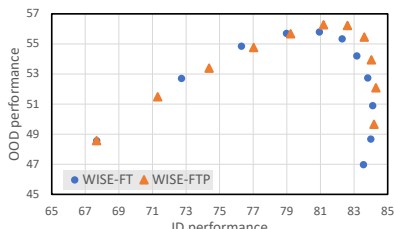

Figure 3: ImageNet WISE Interpolation [8] Result using CLIP ViT-Base Fine-tuned models.

Table 3: ImageNet Fine-tuning Result using CLIP ViT-Base.

| | ID | OOD | | | | Statistics | |
|---|---|---|---|---|---|---|---|
| | Im | ImV2 | Im-A | Im-R | Im-S | OOD Ave. | Ave. |
| zero-shot | 67.68 | 61.41 | 30.60 | 56.77 | 45.53 | 48.58 | 52.40 |
| vanilla FT | 83.66 | 73.82 | 21.40 | 43.06 | 45.52 | 46.98 | 54.29 |
| Linear Prob. | 78.25 | 67.68 | 26.54 | 52.57 | 48.26 | 48.76 | 54.66 |
| LP-FT [7] | 82.99 | 72.96 | 21.08 | 44.65 | 47.56 | 46.56 | 53.85 |
| L2-SP [35] | 83.44 | 73.2 | 20.55 | 43.89 | 46.60 | 46.06 | 53.54 |
| FTP | **84.19** | **74.64** | 26.50 | 47.23 | 50.23 | **49.65** | **56.56** |
| WISE-FT [8] | 80.94 | 72.47 | 33.18 | **63.33** | 54.20 | 55.58 | 60.82 |
| WISE-FTP | **82.61** | **74.09** | **34.56** | 61.18 | **55.06** | **56.22** | **61.50** |

However, there are two limitations: 1) not all pre-trained models have this property of *linear connectivity* and 2) a zero-shot classifier head is needed to initialize the linear classifier head. *Our contribution is orthogonal to WISE because FTP is a general optimization algorithm whereas WISE is a post-training algorithm for specific zero-shot models.* Therefore, we first compare FTP to vanilla fine-tuning and then apply WISE to both models. We follow the public code base of DEIT [39] to train our CLIP pre-trained ViT-Base. Specifically, we use weight-decay (0.1), drop-path (0.2) [40], label-smoothing (0.1) [41], Mixup (0.8) [42] and Cutmix (1.0) [43]. We train our model on ImageNet and report OOD performance on ImageNet-V2 [44], ImageNet-A [45], ImageNet-R [46], and ImageNet-S [47]. Please refer to Appendix 8.4 for more details on implementation.

**FTP outperforms vanilla fine-tuning and improves WISE performance.** In Tab. 3, we report performance for competing methods. Even with various regularizations and augmentations in place, FTP can further improve ID performance on ImageNet. Furthermore, FTP brings better OOD performance on all four OOD datasets. This shows that FTP successfully maintains the robustness of the pre-trained CLIP model while existing regularization such as weight decay and drop-path do not. We also report the interpolation results using WISE [8] for the vanilla fine-tuned and FTP fine-tuned models. We sweep a range of interpolation ratios $\in \{0.1, 0.2, ..., 0.9\}$ and show the trajectory of ID vs. OOD performance plot in Fig. 3. The models with the best average performance are reported in the lower portion of Tab. 3. As expected, WISE interpolation significantly improves OOD generalization for both methods. However, WISE-FTP has significantly better ID performance while still having better OOD performance. This shows that improvement to the base fine-tuning strategy can further benefit pose-training methods such as WISE.

## 4.2 PASCAL Dense Vision Task Experiments

Table 4: Pascal Semantic Segmentation Results using SWIN-Tiny transformers pre-trained on ImageNet21K. Performance is measured by mIoU↑. FTP achieves the best OOD performance and is much faster than prior work TPGM [10] by **34**%.

| | ID | OOD | | | | Statistics | | | |
| | Clean | Fog | Defocus | Gaussian | Brightness | OOD Avg. | ID Δ (%) | OOD Δ (%) | Time (s/it)↓ |
|---|---|---|---|---|---|---|---|---|---|
| Vanilla FT | 66.03 (0.37) | 56.72 (0.83) | 38.04 (0.83) | 23.21 (0.96) | 58.03 (0.66) | 44.00 | 0.00 | 0.00 | 0.288 |
| Adapter [24] | 71.85 (0.06) | 69.36 (0.07) | 50.94 (0.25) | 37.43 (0.64) | 68.26 (0.08) | 56.50 | 8.82 | 28.40 | 0.233 |
| BitFit [30] | 70.31 (0.11) | 67.00 (0.24) | 46.39 (0.35) | 30.61 (0.51) | 66.22 (0.16) | 52.56 | 6.49 | 19.44 | 0.248 |
| L2-SP [35] | 73.47 (0.06) | 69.87 (0.04) | 49.20 (0.43) | 39.10 (0.84) | 68.61 (0.24) | 56.70 | 11.27 | 28.85 | 0.347 |
| MARS-SP [9] | 66.24 (0.23) | 56.97 (0.79) | 37.29 (1.20) | 21.82 (2.06) | 58.27 (0.33) | 43.59 | 0.32 | -0.94 | 0.318 |
| LLRD [48] | 72.09 (0.06) | 68.13 (0.25) | 46.18 (1.30) | 37.28 (2.54) | 66.30 (0.29) | 54.47 | 9.18 | 23.79 | 0.289 |
| TPGM [10] | 72.56 (0.06) | 69.51 (0.57) | 50.88 (0.97) | 38.62 (1.04) | 68.82 (0.25) | 56.96 | 9.89 | 29.44 | 0.611 |
| FTP | **73.79** (0.10) | **71.10** (0.23) | **52.63** (0.75) | **40.25** (0.21) | **69.81** (0.49) | **58.45** | **11.76** | **32.83** | 0.401 |

To further demonstrate the effectiveness of FTP in more diverse scenarios, we test it on PASCAL-Context [49]. Specifically, following the prior work [50], we use the PASCAL-Context datasets [49], which consist of labels for semantic segmentation, human parts segmentation, and surface normal estimation. For OOD performance, following the popular natural robustness literature [51], we report results on various degradations including fog, defocus blur, Gaussian noise, and brightness corruption, with 5 severity each. We use a combination of Swin ViT-Tiny [52] (pre-trained on ImageNet-22K) and Segformer [53]. In this architecture, Swin Transformer serves as the feature extraction backbone and Segformer is the task-specific decoder. While the feature backbone is initialized with pre-trained weights, a significant part of the entire model (the Segformer decoder) is randomly initialized; In contrast, in simple classification (Sec. 4.1.1), only the last linear classification layer is randomly initialized. Please refer to Appendix 8.5 for details.

**FTP achieves the best ID performance and OOD generalization.** We report results for semantic segmentation, human parts segmentation, and surface normal estimation in Tab. 4, Appendix Tab. 7, and Appendix Tab. 8 respectively. We additionally add Layer-Wise Learning Rate decay [48] ( LLRD) as a strong baseline. Notably, in all three tasks, FTP outperforms vanilla fine-tuning on ID performance by 11.71%, 4.48%, and 18.30% respectively. This demonstrates the effectiveness of projection as a regularization technique for transfer learning. More importantly, the OOD performance improves as large as 33.02% in semantic segmentation. This shows that 1) FTP can effectively maintain the robustness of the original pre-trained model; 2) even though the entire decoder component is randomly initialized, it is worthwhile to put regularizations on the pre-trained feature backbone.

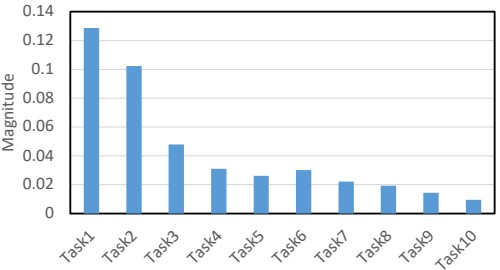

Figure 4: Average Learned Constraints for each task using FTP.

Table 5: CL Results on ImageNet-R

| Method | $A_{1:N}$ ($\uparrow$) | $F_N$ ($\downarrow$) |
|---|---|---|
| FT++ [54] | $48.93 \pm 1.15$ | $9.81 \pm 0.31$ |
| LwF.MC [55] | $66.73 \pm 1.25$ | $3.52 \pm 0.39$ |
| L2P++ [56] | $71.66 \pm 0.64$ | $1.78 \pm 0.16$ |
| DualPrompt [57] | $71.32 \pm 0.62$ | $1.71 \pm 0.24$ |
| CODA-P [54] | $75.45 \pm 0.56$ | $1.64 \pm 0.10$ |
| EWC [58] | $64.66 \pm 2.04$ | $1.55 \pm 0.25$ |
| L2 [54] | $76.06 \pm 0.65$ | $1.68 \pm 0.16$ |
| FTP | $76.06 \pm 0.35$ | $2.27 \pm 0.18$ |
| FTP + EWC | $\mathbf{77.26 \pm 0.40}$ | $\mathbf{1.48 \pm 0.15}$ |

## 4.3 Continual Learning (CL) Experiments

Recently, pre-trained models have been shown to greatly improve the performance of CL algorithms [59]. We follow the settings in this work [59] to partition ImageNet-R (200 classes) into 10 sequential tasks with 20 non-overlapping classes in each task. A model is trained on each task only once sequentially. To use FTP for CL tasks, unlike supervised vision tasks (Sec. 4.1, 4.2), we re-initialize FTP after each task and use the *current* model as the "pre-trained model" for the next task. Moreover, inspired by the prior work [59], we use FTP to only fine-tune the attention blocks. We report both the final task accuracy across all tasks $A_{1:N} \uparrow$ and the global forgetting $F_N \downarrow$ in Tab. 5 to analyze *plasticity* and *forgetting*. Please refer to Appendix 8.6 for more on the metrics and experimental setup. In Table 5, we benchmark against the popular and recent rehearsal-free continual learning methods. FTP alone achieves state of art accuracy against all methods and relatively good forgetting compared to vanilla FT, a sign of superior *plasticity* and balanced *forgetting*. We visualize the learned constraints for each task in Fig. 4. We observe that while each task is independent and FTP is re-initialized each time, FTP learns stronger regularization for later tasks. This contributes to lower forgetting compared to FT. *We found that FTP combined with a simple continual learning method, EWC [58], achieves state-of-the-art in this setting.* Compared to the prompting methods L2P, DualPrompt, and the recent CODA-Prompt, FTP has clear and significant improvements. Our intuition is that the combination of the superior plasticity of FTP and the low forgetting of EWC is the key to the improvement.

## 5 Limitations

Like any regularization method, FTP has a hyper-parameter to adjust its regularization strength. In this case, the positive gradient annealing factor $0 \leq \kappa \leq 1$ (default 1) (Sec. 3.2) controls the strength of projection with smaller values indicating weaker regularization. Note that $\kappa = 0$ means that the projection constraints are *non-decreasing* during training. In this case, FTP still provides regularization. For example, we found that a $\kappa = 0$ is necessary to obtain the best performance for some dense vision tasks in Appendix 8.5. Generally, we recommend starting with the default $\kappa$ and only tuning it if underfitting is observed.

## 6 Conclusion

In this paper, we proposed Fast Trainable Projection, a fine-tuning algorithm to maintain the robustness and the generalization capability of the pre-trained model. FTP learns projection constraints for each layer in a neural network efficiently by carefully re-using past information to save computation. To understand the connection between robustness and projection, we provided a holistic discussion of fine-tuning robustness from its feature space definition to the weight space dual. The new perspective lends a mathematical foundation to the idea of using projection in fine-tuning. Across four vision tasks with different pre-trained models, FTP demonstrated superior ID and OOD generalization capability and significantly better computation efficiency. Furthermore, the continual learning experiments demonstrated FTP's potential in other deep learning paradigms beyond simple fine-tuning. Combined with its compatibility with popular optimization algorithms, we believe FTP can be broadly beneficial in improving the performance of learning tasks using pre-trained initialization.

# 7 Acknowledgements

This work was supported by ONR grant N00014-18-1-2829.

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

# 8 Appendix

## 8.1 Proof of Lemma 1

Prior works [9, 10] have used the high-level notion that staying "close" to the pre-trained model can help maintain its robustness capability to justify using projection for fine-tuning. However, there is more than one way to encourage this, for example, regularization [35], a small learning rate [7], and projection [10]. It is not immediately clear why projection is a principled approach. To understand FTP's capability to maintain the pre-trained mode's robustness, we first propose to establish a connection between Lipschitz continuity, a commonly used measure of robustness [19, 20, 21], and fine-tuning through a new definition of *difference function* in the Lemma 1.

*Proof.* We first expand the difference functions in Eq. 9, i.e. plugging in $\Delta h(\cdot) = h_f(\cdot) - h_0(\cdot)$,

$$\|\Delta h(\mathbf{x}) - \Delta h(\mathbf{x}')\|_{\mathrm{h}} \leq L_d \|\mathbf{x} - \mathbf{x}'\|_{\mathrm{x}} \quad \forall (\mathbf{x}, \mathbf{x}') \in \mathbb{R}^m \tag{11}$$
$$\rightarrow \| (h_f(\mathbf{x}) - h_0(\mathbf{x})) - (h_f(\mathbf{x}') - h_0(\mathbf{x}')) \|_{\mathrm{h}} \leq L_d \|\mathbf{x} - \mathbf{x}'\|_{\mathrm{x}}$$
$$\rightarrow \| (h_f(\mathbf{x}) - h_f(\mathbf{x}')) - (h_0(\mathbf{x}) - h_0(\mathbf{x}')) \|_{\mathrm{h}} \leq L_d \|\mathbf{x} - \mathbf{x}'\|_{\mathrm{x}}$$

Then we apply the reverse triangular inequality to the left-hand side of Eq. 11.

$$|\|h_f(\mathbf{x}) - h_f(\mathbf{x}')\|_{\mathrm{h}} - \|h_0(\mathbf{x}) - h_0(\mathbf{x}')\|_{\mathrm{h}}| \leq \| (h_f(\mathbf{x}) - h_f(\mathbf{x}')) - (h_0(\mathbf{x}) - h_0(\mathbf{x}')) \|_{\mathrm{h}}$$

Therefore, we have,

$$\|h_f(\mathbf{x}) - h_f(\mathbf{x}')\|_{\mathrm{h}} - \|h_0(\mathbf{x}) - h_0(\mathbf{x}')\|_{\mathrm{h}} \leq L_d \|\mathbf{x} - \mathbf{x}'\|_{\mathrm{x}} \tag{12}$$
$$\rightarrow \|h_f(\mathbf{x}) - h_f(\mathbf{x}')\|_{\mathrm{h}} \leq L_d \|\mathbf{x} - \mathbf{x}'\|_{\mathrm{x}} + \|h_0(\mathbf{x}) - h_0(\mathbf{x}')\|_{\mathrm{h}}$$

Assuming that the pre-trained model $h_0$ is $L_0$-Lipschitz, we know that $\|h_0(\mathbf{x}) - h_0(\mathbf{x}')\|_{\mathrm{h}} \leq L_0 \|\mathbf{x} - \mathbf{x}'\|_{\mathrm{x}}, \quad \forall (\mathbf{x}, \mathbf{x}') \in \mathbb{R}^m$. Plug this into Eq. 12,

$$\|h_f(\mathbf{x}) - h_f(\mathbf{x}')\|_{\mathrm{h}} \leq (L_d + L_0) \|\mathbf{x} - \mathbf{x}'\|_{\mathrm{x}} \tag{13}$$

$\square$

## 8.2 Proof of Lemma 2

In the previous section, we established a connection between the robustness of a fine-tuned model $h_f(\cdot)$ and its difference function $\Delta h(\cdot)$. Naturally, if we can limit the Lipschitz constant $L_d$ of the difference function, we can maintain the robustness of the pre-trained model. In this section, we show *projection* as an effective method to enforce the $L_d$-Lipschitz condition in Eq 9.

*Proof.* **Linear Operators.** A neural network is composed of linear operators with connecting non-linear activations. Following prior works [9, 10], we analyze the linear operators[3]: $h(\mathbf{x}) = \mathbf{W}\mathbf{x} + \mathbf{b}, \mathbf{W} \in \mathbb{R}^{n \times m}, \mathbf{b} \in \mathbb{R}^n$. Let's define $h_f(\mathbf{x}) = \mathbf{W}_f \mathbf{x} + \mathbf{b}_f$ and $h_0(\mathbf{x}) = \mathbf{W}_0 \mathbf{x} + \mathbf{b}_0$, and plug them in Eq. 9.

$$\|(\mathbf{W}_f - \mathbf{W}_0)(\mathbf{x} - \mathbf{x}')\|_{\mathrm{h}} \leq L_d \|\mathbf{x} - \mathbf{x}'\|_{\mathrm{x}} \quad \forall (\mathbf{x}, \mathbf{x}') \in \mathbb{R}^m.$$

Rearranging the above equation gives us an upper bound on $L_d$,

$$L_d = \sup \left\{ \frac{\|(\mathbf{W}_f - \mathbf{W}_0)(\mathbf{x} - \mathbf{x}')\|_{\mathrm{h}}}{\|\mathbf{x} - \mathbf{x}'\|_{\mathrm{x}}} \quad \forall (\mathbf{x}, \mathbf{x}') \in \mathbb{R}^m \right\}. \tag{14}$$

**Matrix Norms.** Eq. 14 matches the definition of a matrix norm for a matrix $\mathbf{W} \in \mathbb{R}^{n \times m}$: $\|\mathbf{W}\|_{\mathrm{h,x}} = \sup \left\{ \frac{\|\mathbf{W}x\|_{\mathrm{h}}}{\|x\|_{\mathrm{x}}}, \quad \forall x \in \mathbb{R}^n \quad \text{with} \quad x \neq 0 \right\}.$ *Therefore, to minimize $L_d$ in Eq. 9, we just need to minimize the matrix norm $\|\mathbf{W}_f - \mathbf{W}_0\|_{\mathrm{h,x}}$.* Note that different vector norm combinations ($\| \cdot \|_{\mathrm{h}}$ and $\| \cdot \|_{\mathrm{x}}$) will lead to a different matrix norm $\| \cdot \|_{\mathrm{h,x}}$. Certain vector norm combinations have a closed-form matrix norm while the majority do not. Following prior works [9, 10], we use Maximum Absolute Row Sum (MARS) matrix norm, which is characterized by $l_\infty$ vector norms in both domains.

---

[3]Convolutional layers can be also written in the matrix multiplication form using Toeplitz matrix.

Specifically, given a desired constraint $L_d$, we want $\|\mathbf{W}_f - \mathbf{W}_0\|_{\infty,\infty} \le L_d$. Per the definition of the MARS matrix norm, which is the largest $l_1$ norm of each row of a matrix, the inequality can be equivalently enforced for each row independently, i.e.,

$$\|\mathbf{W}_f - \mathbf{W}_0\|_{\infty,\infty} \le L_d \iff \|\mathbf{w}_f^i - \mathbf{w}_0^i\|_1 \le L_d, \quad \forall i \in \{1, ..., n\}. \tag{15}$$

where $\mathbf{w}^i$ denotes the $i$-th row of the matrix $\mathbf{W}$.

**Projection.** To ensure the inequality in Eq. 15, we can *project* $\mathbf{W}_f$ towards $\mathbf{W}_0$ using the following projection equation. For each row $\mathbf{w}^i$ in a matrix $\mathbf{W}$, the projected weight $\tilde{\mathbf{w}}_p^i$ is calculated by

$$\mathbf{w}_p^i = \min\left(1, \frac{\gamma}{\|\mathbf{w}_f^i - \mathbf{w}_0^i\|_1}\right)(\mathbf{w}_f^i - \mathbf{w}_0^i) + \mathbf{w}_0^i.$$

It is easy to check that $\mathbf{w}_p^i$ satisfies Eq. 15, i.e., $\|\mathbf{w}_p^i - \mathbf{w}_0^i\|_1 \le L_d$ if $0 \le \gamma \le L_d$. $\qquad\square$

**Lipschitz Bound.** Since a neural network is a composition of linear operators and non-linear activations, by the composition rule of the Lipschitz functions, an upper bound of the entire network is just the product of the Lipschitz constant for each linear operator and non-linear activations, where most non-linear activations are 1-Lipschitz [21]. However, the Lipschitz bound obtained by using the composition rule is not a tight bound on the entire network. While it is an active research area to find tighter bounds for neural networks without relying on the layer-wise composition rule [60, 20], the layer-wise approach is particularly suitable for connecting the fine-tuning process and Lipschitz continuity because it leads to layer-wise regularization techniques as we demonstrated above.

## 8.3 FTP: Additional Discussion

In the main paper Sec. 3.2, we described the algorithmic difference between TPGM and FTP. However, there is an implicit assumption made as a result of the difference. We now discuss the implications of it. After obtaining the updated constraints $\gamma_t$ in Eq. 5, if the algorithm were to follow TPGM, the next step would be applying the updated constraints to *re-calculate* the previous model $\mathbf{W}_{t-1}$. However, instead of rolling back, FTP applies the updated constraints directly to the current unconstrained model $\tilde{\mathbf{W}}_t$. This step assumes *smoothness* in the update of $\gamma_t$, i.e., the $\gamma_t$ does not change drastically in consecutive steps. The assumption is valid since $\gamma_t$ is updated by *AdampUpdate* (Alg. 2 below) which uses a moving average update with a momentum of 0.9. So the change of $\gamma_t$ is very smooth because of the high discount factor of 0.9. Importantly, we have **re-used** the same gradient $\mathbf{g}_t$ available for computing the current unconstrained model $\tilde{\mathbf{W}}_t$. This is the key to saving computation because calculating the forward and backward pass through the model is the main computation bottleneck in TPGM because it requires a separate training loop as a result of "rolling back".

---

**Algorithm 2** AdampUpdate: AdamUpdate implements one step update of Adam [38]

---

**Require:** $\gamma_{t-1}, \nabla\gamma_t, t$ $\hfill \triangleright$ Input
**Require:** $\mu \leftarrow 1e-2, (\beta_1, \beta_2) \leftarrow (0.9, 0.999), \epsilon \leftarrow 1e-8$ $\quad\triangleright$ Fixed parameters for AdamUpdate
**Require:** $m_1 \leftarrow 0$ $\hfill \triangleright$ Initialize 1st moment vector
**Require:** $v_1 \leftarrow 0$ $\hfill \triangleright$ Initialize 2nd moment vector
$\quad m_t \leftarrow \beta_1 m_{t-1} + (1-\beta_1)\nabla\gamma_t$
$\quad v_t \leftarrow \beta_2 v_{t-1} + (1-\beta_2)\nabla\gamma_t^2$
$\quad \hat{m}_t \leftarrow m_t/(1-\beta_1^t)$
$\quad \hat{v}_t \leftarrow v_t/(1-\beta_2^t)$
$\quad \gamma_t \leftarrow \gamma_{t-1} - \mu\hat{m}_t/(\sqrt{\hat{v}_t} + \epsilon)$

---

## 8.4 Image Classification Experiments Details and Additional Results

In Sec. 4.1.1, we presented image classification results on DomainNet-100% data (111,307 images). Now we further present results using only 10% (11,031 images) of the training data in Tab. 6. In this case, projection-based methods, TPGM and FTP achieved the best performance, demonstrating their regularization capability under low-label conditions. Similar to findings in the main paper, FTP is

Table 6: **DomainNet Results using CLIP pre-trained ResNet50 with 10% Real Data.** FFTP achieves competitive OOD performance and is much faster than prior work TPGM [10] by 37%.

| | ID | OOD | | | | Statistics | | | |
| | Real | Sketch | Painting | Infograph | Clipart | OOD Avg. | ID Δ (%) | OOD Δ (%) | Time (s/it)↓ |
|---|---|---|---|---|---|---|---|---|---|
| Vanilla FT | 57.35 (1.43) | 17.48 (0.68) | 25.60 (0.70) | 10.30 (1.57) | 23.01 (0.65) | 19.10 | 0.00 | 0.00 | 0.54 |
| LP | 47.19 (0.93) | 17.81 (0.25) | 22.71 (2.08) | 17.13 (0.75) | 17.59 (0.69) | 18.81 | -17.71 | -1.52 | 0.13 |
| PF [32] | 71.04 (0.91) | 27.87 (1.04) | 38.31(1.05) | 19.85 (0.70) | **33.92** (1.53) | 29.99 | 23.86 | 57.01 | 0.31 |
| L2-SP [35] | 61.41 (0.92) | 22.61 (0.52) | 30.48 (0.42) | 12.28 (0.50) | 26.59 (0.57) | 22.99 | 7.08 | 20.37 | 0.61 |
| MARS-SP [9] | 52.53 (0.84) | 15.34 (0.54) | 21.57 (0.45) | 8.49 (0.60) | 19.96 (0.01) | 16.34 | -8.41 | -14.44 | 0.60 |
| LP-FT [7] | 64.11 (0.78) | 20.54 (0.27) | 30.89 (0.41) | 13.58 (0.63) | 29.55 (0.82) | 23.64 | 11.78 | 23.77 | - |
| TPGM [10] | **73.16** (1.27) | **29.88** (0.81) | 36.80 (1.42) | 19.72 (0.12) | 35.28 (0.74) | **30.42** | 27.56 | 59.27 | 1.10 |
| FTP | 72.89 (0.34) | 27.44 (0.13) | **38.11** (0.26) | **20.20** (0.26) | 33.58 (0.49) | 29.83 | 27.10 | 56.19 | 0.69 |

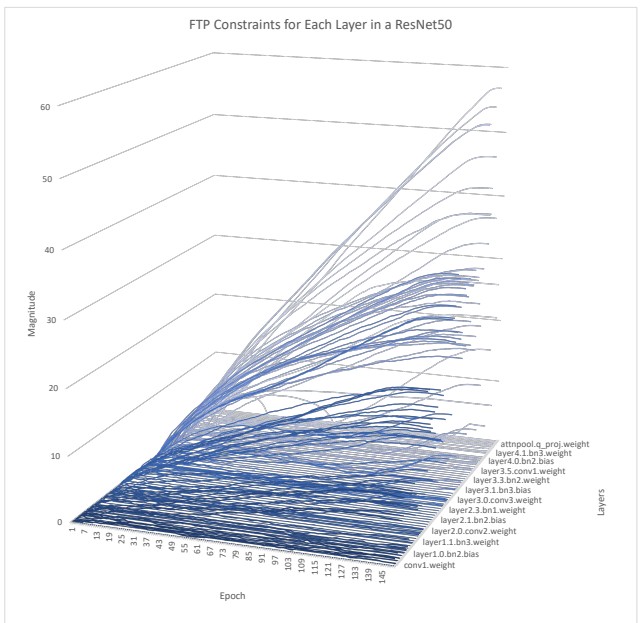

Figure 5: Visualization of learned FTP constraints. **Settings:** We fine-tune a pre-trained ResNet50 on DomainNet-Real for 150 epochs. There are in total 174 constraints imposed on the model excluding the last linear layer. **Observations:** 1) Early layers (dark colors) generally have smaller constraints than the latter layers (light colors) throughout training. 2) Constraints grow from small to large and converge in the end.

up to 37% faster than TPGM during training. Next, we describe the hyper-parameters for all image classification experiments in Sec. 4.1 and above.

**DomainNet.** We use the released code from the prior work, TPGM [10] to train our FTP model. Therefore, we directly use the reported results from TPGM for competing methods. For FTP, we apply constraints to all trainable layers except for the last linear classification layers. For all experiments, we use SGD as the base optimizer with a weight decay of $5e-4$. For DomainNet-100% and DomainNet-10% experiments, we train models for 50 and 150 epochs respectively with a batch size of 256. We sweep a range of learning rates and use the validation split to determine the best learning rate for FTP for each experiment. Here is the list of best-validated learning rates for all DomainNet experiments. We also provide a visualization of the learned constraints in Fig. 1b.

- DomainNet-100% MOCO-V3 ResNet50 (Tab. 1): $1e-2$
- DomainNet-100% CLIP ResNet50 (Tab. 2): $1e-2$
- DomainNet-10% CLIP ResNet50 (Tab. 6): $1e-1$

Note that we use the default $\kappa=1$ for all these experiments. Every DomainNet experiment was conducted using 4 RTX 2080 GPUs.

**ImageNet.** For ImageNet experiments (Tab. 3, Fig. 3), we use a CLIP pre-trained ViT-Base [4]. Unlike the DomainNet experiments, we also initialize the last linear layer with zero-shot weights extracted from a CLIP text encoder, following the prior work WISE [8]. Therefore, FTP is applied to all trainable layers including the last linear layer. Training Transformers have been well-studied with abundant regularization and augmentation techniques. To obtain the best fine-tuning performance, we follow the public code base of DEIT [39] to fine-tune all methods. Specifically, we use weight-decay (0.1), drop-path (0.2) [40], label-smoothing (0.1) [41], Mixup (0.8) [42] and Cutmix (1.0) [43]. One exception is Linear-Probing (LP), where we do not use any of the above augmentations because they have been shown to degrade linear probing performance [3, 1]. We train all methods using AdamW [37] as the base optimizer with a weight decay of 0.1, cosine learning rate schedule, and a batch size of 256 for 30 epochs. We also sweep relevant hyper-parameters for each method and document them below.

- FT: learning rate $2e - 5$
- LP: learning rate $5e - 3$
- LP-FT: learning rate $2e - 5$. We take the best LP model (trained for 30 epochs) and then fine-tune it for another 15 epochs with the learning rate specified above.
- L2-SP: learning rate $2e - 5$, regularization hyper-parameter $1e - 5$.
- FTP: learning rate $3e - 5$, regularization hyper-parameter default $\kappa = 1$.

Every ImageNet classification experiment was conducted on 2 A40 GPUs.

## 8.5  PASCAL Dense Vision Task Experiments Details and Additional Results

In Sec. 4.2, we presented results on semantic segmentation. In this section, we provide the additional results on semantic segmentation and surface normal estimation in Tab, 7 and Tab. 8. FTP achieves the best ID and OOD performance with significantly improved computation efficiency over TPGM [10]. Next, we will give more details on implementation.

Table 7: Pascal Human Parts Segmentation Results using SWIN-Tiny transformers pre-trained on ImageNet21K. Performance is measured by mIoU↑. FTP achieves the best OOD performance and is much faster than prior work TPGM [10] by **34**%.

|  | ID | OOD | | | | | Statistics | | |
|---|---|---|---|---|---|---|---|---|---|
|  | Clean | Fog | Defocus | Gaussian | Brightness | OOD Avg. | ID Δ (%) | OOD Δ (%) | Time (s/it)↓ |
| Vanilla FT | 62.61 (0.31) | 57.50 (0.73) | 40.76 (0.19) | 30.64 (0.88) | 57.47 (0.33) | 46.59 | 0.00 | 0.00 | 0.280 |
| Adapter | 60.84 (1.27) | 57.11 (0.39) | 45.03 (3.96) | 33.12 (1.92) | 57.25 (0.68) | 48.13 | -2.81 | 3.30 | 0.221 |
| BitFit | 59.06 (0.97) | 55.66 (1.36) | **45.81** (1.27) | 32.18 (2.59) | 55.89 (0.97) | 47.39 | -5.67 | 1.70 | 0.235 |
| L2-SP | 62.26 (3.17) | 58.46 (2.83) | 45.35 (1.30) | 34.36 (2.79) | 58.40 (2.52) | 49.14 | -0.56 | 5.47 | 0.336 |
| MARS-SP | 62.92 (0.94) | 58.04 (1.75) | 42.51 (1.72) | 32.66 (2.53) | 58.33 (1.15) | 47.89 | 0.50 | 2.77 | 0.308 |
| LLRD | 64.37 (1.80) | 60.10 (2.58) | 44.61 (1.95) | 36.90 (4.84) | 59.84 (2.06) | 50.36 | 2.81 | 8.09 | 0.278 |
| TPGM | 63.29 (1.72) | 60.16 (1.44) | 46.91 (1.78) | 37.30 (2.60) | 59.81 (1.00) | 51.04 | 1.10 | 9.55 | 0.602 |
| FTP | **65.50** (0.17) | **61.73** (0.36) | 44.97 (0.70) | **40.55** (1.71) | **61.23** (0.12) | **52.12** | **4.63** | **11.86** | 0.397 |

Table 8: Pascal surface normal Results using SWIN-Tiny transformers pre-trained on ImageNet21K. Performance is measured by RMSE↓. FTP achieves the best OOD performance and is much faster than prior work TPGM [10] by 35%.

|  | ID | OOD | | | | | Statistics | | |
|---|---|---|---|---|---|---|---|---|---|
|  | Clean | Fog | Defocus | Gaussian | Brightness | OOD Avg. | ID Δ (%) | OOD Δ (%) | Time (s/it)↓ |
| Vanilla FT | 18.98 (0.05) | 22.25 (0.08) | 23.51 (0.06) | 27.33 (0.20) | 20.83 (0.06) | 23.48 | 0.00 | 0.00 | 0.288 |
| Adapter | 18.19 (0.05) | 20.15 (0.04) | 21.46 (0.02) | 23.90 (0.14) | 19.23 (0.06) | 21.19 | -4.15 | -9.77 | 0.229 |
| BitFit | 20.01 (0.05) | 21.93 (0.03) | 23.95 (0.12) | 26.92 (0.18) | 21.28 (0.05) | 23.52 | 5.43 | 0.17 | 0.240 |
| L2-SP | 16.51 (0.04) | 19.26 (0.13) | 20.49 (0.11) | 24.46 (0.29) | 18.08 (0.04) | 20.57 | -13.01 | -12.38 | 0.343 |
| MARS-SP | 19.01 (0.04) | 22.15 (0.13) | 23.69 (0.11) | 27.53 (0.29) | 20.86 (0.04) | 23.56 | 0.18 | 0.32 | 0.313 |
| LLRD | 15.54 (0.08) | 18.31 (0.03) | **20.01** (0.20) | 26.47 (1.45) | 17.36 (0.07) | 20.54 | -18.11 | -12.54 | 0.279 |
| TPGM | 18.17 (0.02) | 19.74 (0.04) | 21.00 (0.15) | **23.53** (0.27) | 19.02 (0.03) | 20.82 | -4.24 | -11.32 | 0.616 |
| FTP | **15.51** (0.10) | **18.19** (0.09) | **20.01** (0.21) | 26.39 (0.78) | **17.32** (0.10) | **20.48** | **-18.30** | **-12.79** | 0.403 |

Following prior works [50], we use a combination of Swin-Tiny Transformer [52] encoder and Segformer [53] decoder. The decoder is customized to allow different output formats. Only the Swin

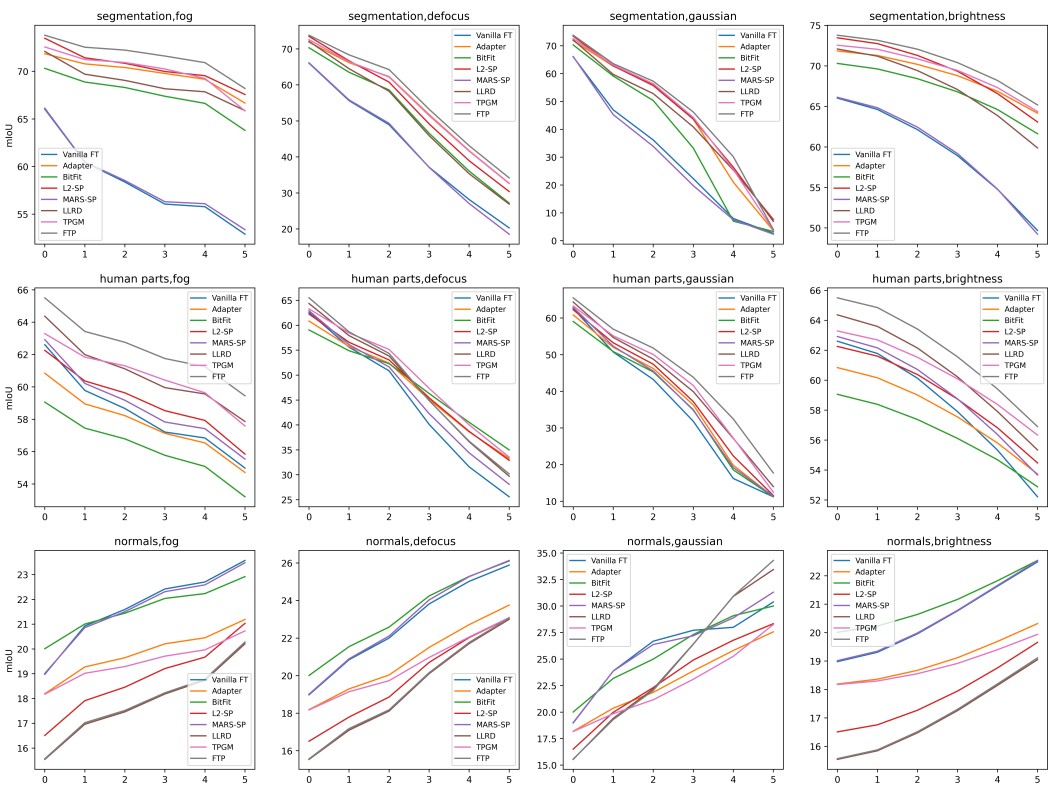

Figure 6: Performance Breakdown for each Level of Corruption on PASCAL-Context Vision Tasks.

encoder is initialized with pre-trained weights (pre-trained on ImageNet-22k). Therefore, we only apply FTP to the encoder. For all methods, we use Adam as the base optimizer with a weight decay of $1e-4$ and a learning rate of $1e-4$ for 60 epochs. For methods with regularization hyper-parameters, we sweep a range of values and report the best one. We provide Tab. 9 for reference.

Table 9: Hyper-parameters for PASCAL Dense Vision Tasks Experiments.

|          | Semseg | Human Parts | Surface Normal |
|----------|--------|-------------|----------------|
| L2-SP    | 5e-4   | 1e-4        | 1e-4           |
| LLRD     | 0.65   | 0.45        | 0.65           |
| MARS-SP  | 4      | 8           | 4              |
| FTP      | 1.0    | 0.0         | 0.0            |

To test OOD robustness on the PASCAL-Context benchmark, we apply natural corruptions to the original clean images. Specifically, we select four types of corruptions from the popular benchmark [51], each of which is sampled from a main category: noise, blur, weather, and digital. Each corruption has five levels of severity. We report the average values over the five severity in our paper. Here, we also provide a detailed breakdown for each level of corruption in Fig. 6. Every PASCAL experiment was conducted on a single RTX 2080 GPU.

## 8.6 Continual Learning Experiments Details and Additional Results

In this section, we provide a brief overview of the settings in continual learning (CL). In CL, a model $\theta$ is trained on a sequence of task $n \in \{1, ..., N\}$. Each task has a non-overlapping set of class labels $T_n$, and we denote the number of classes as $|T_n|$. For ImageNet-R, we split the 200 classes into 10 tasks with 20 labels each, i.e., $N = 10, |T_n| = 20$. Our experiments belong to the class-incremental category in CL. With each new task, the final linear classifier layer is expanded

with randomly initialized weights. We denote $\theta_{i,1:n}$ as the model that has been trained on $i$ tasks and the classifier has all classes up to and including the $n$-th task ($i \geq n$).

To measure global performance, we first define the *global task accuracy* $A_{1:N}$ as,

$$A_{1:N} = \frac{1}{|D^{\text{test}}|} \sum_{(x,y) \in D^{\text{test}}} \mathbf{I}(\hat{y}(x, \theta_{N,1:N}) = y).$$

where $D_{\text{test}}$ is the test dataset which has data from all $N$ tasks and $\hat{y}(x, \theta)$ denotes the predicted class from the model with weights $\theta$. Then we define the *global forgetting $F_N$* [61] as,

$$F_N = \frac{1}{N-1} \sum_{i=2}^{N} \sum_{n=1}^{i-1} \frac{|T_N|}{T_{1:n}} (R_{n,n} - R_{i,n})$$

where,

$$R_{i,n} = \frac{1}{|D_n^{\text{test}}|} \sum_{(x,y) \in D_n^{\text{test}}} \mathbf{I}(\hat{y}(x, \theta_{i,1:n}) = y).$$

Following the prior work [59], all experiments in Tab. 5 use a ViT-Base pre-trained on ImageNet. We tune FTP with the code provided by the authors and directly compare it to the results from the prior work. Specifically, all methods use Adam as the base optimizer with no weight decay and a batch size of 128. All results are averaged over 3 random seed trials where the class allocation to each task is shuffled. For FTP, we train the model for 25 epochs with an initial learning rate of $5e - 4$ and a cosine learning rate schedule. For all methods, we freeze the majority of the backbones and only fine-tune the QKV attention layers in the ViT. Please refer to the prior work for a more detailed description of the compared methods. Every CL experiment was conducted on 4 RTX2080 GPUs.

## 8.7 Pytorch Code Example of FTP

Here is an example of using SGDP (SGD+FTP) in Pytorch format. SGDP requires the common arguments for initializing an SGD optimizer class in Pytorch with two additional inputs: $k$ and exclude_set. $k$ is the hyper-parameter for positive gradient annealing (Sec. 3.2) and exclude_set contains the set of the names of parameters to be excluded from the projection operation. A complete demonstration of image classification is provided in the supplementary. You should be able to reproduce FTP results in Tab. 1 and Tab. 2.

```python
from FTP import SGDP

# Parameters to be optimized
params_to_opt = [x[1] for x in model.named_parameters()]
# Names of parameters to be optimized
params_to_opt_name = [x[0] for x in model.named_parameters()]
# Copy the initial parameters as the anchor
params_anchor = copy.deepcopy(params_to_opt)
# Set up the parameter groups
param_group = [{"params":params_to_opt,
                "pre": params_anchor,
                "name": params_to_opt_name}]
# Set up the optimization hyper-parameters
optimizer_params = {
        "lr": 1e-2,
        "weight_decay": 5.0e-4,
        "momentum": 0.9,
        "nesterov": True,
        "k":1.0,
        "exclude_set":{"module.head.weight","module.head.bias"}
    }
optimizer = SGDP(param_group,**optimizer_params)
```

