# OpenReview forum: "Fast Trainable Projection for Robust Fine-tuning"
_NeurIPS.cc/2023/Conference — NeurIPS 2023 poster_

### Official Review · Reviewer_yjP3 · 2023-07-05

**Soundness:** 3 good
**Presentation:** 3 good
**Contribution:** 3 good
**Rating:** 5
**Confidence:** 5

**Summary:**

This paper proposes a robust fine-tuning technique that has the decent scalability and efficiency to achieve higher ID and OOD performance when transferring a pre-trained model to downstream tasks. A new projection-based fine-tuning algorithm, Fast Trainable Projection (FTP) is proposed for computationally efficient learning of per-layer projection constraints with the theoretically analyses in the lens of Lipschitz continuity. Extensive experiments show its effectiveness.

**Strengths:**

The paper is well-written and easy to read.

The proposed FTP is reasonable and easy to implement with the provided pseudo code.

The theoretical analysis is also appreciated.

Extensive experiments on several downstream tasks and datasets are conducted to show the strong performances of the proposed method


**Weaknesses:**

While the method is simple with strong performances, the novelty is somewhat limited. It seems like a direct extension of TPGM. Using the previous model and cached gradients to update the constraints is trivial to me and makes limited technical contributions. Gradient annealing is also like a trick.

Can the theoretical analyses prove that the proposed FTP is more powerful than TPGM? Since the experimental results show the superior performance of FTP.

The ablation study is insufficient. The ablation should be done on ProjUpdate and Gradient Annealing to show the effectiveness of each component.


**Questions:**

I understand FTP is more efficient than TPGM. But why FTP achieves better performance? Could the author give an intuitive explanation?

For the continual learning experiment, the comparison is somewhat not fair, since the proposed method needs to know the task id. (Page 9 line 288 we re-initialize FTP after each task and use the current model as the “pre-trained model” for the next task.)


**Limitations:**

Well addressed.

---

> ### Author Rebuttal · Authors · 2023-08-09
>
> We thank the reviewer for the positive and constructive feedback!
>
> * **Regarding novelty**
>
> The novelty of our method lies in its algorithmic improvement to the prior work TPGM in terms of efficiency, flexibility, and robustness. To successfully use the previous information to update internal parameters in FTP requires rigorous analysis. It requires careful caching of the correct model components from previous steps. For example, in Eq.4, we store previous “unconstrained” update $\tilde{W_{t-1}}$ instead of naively the previous update $W_{t-1}$ for mathematical correctness that’s not immediately clear without analysis. While the end result is a simple algorithm (a strength), our contribution is to come up with this method and rigorously evaluate it, showing significant benefits as other reviewers have mentioned.
>
> The differences in performance can be partially attributed to the algorithmic differences between the two and also see discussion in Appendix 7.3, but we agree that future work could have potential to understanding and designing even better methods.
> In addition to this, the improvement of FTP is also in its efficiency and flexibility.  For example, we demonstrated the regularization strength of FTP in continual learning experiments (sec.4.3). This is not possible with TPGM due to the lack of validation data under a continual learning setting. Hence such differences have practical import.
> Gradient annealing is the most direct way to adjust the regularization strength of FTP much like hyper-parameters in other works.
>
> * **Regarding the theory**
>
>  No, the theory does not indicate that FTP is better than TPGM. The improvement of FTP over TPGM is algorithmic in terms of efficiency and applicability, not theoretical.
> Nevertheless, section 3.3 introduces a general theory on why projection is useful for fine-tuning, a theoretical question that hasn’t been answered in prior works. While you are right that it generally applies to all projection-based methods including FTP and TPGM, we believe it was important to include to progress this area empirically and theoretically. We will add text and organization to explicitly discuss this.
>
> * **Regarding ablation study**
>
> The ProjUpdate component introduces the core update equation in the FTP optimization algorithm. It serves as the mathematical foundation for FTP and cannot be easily replaced by other alternatives without breaking the core mechanism. Therefore, it is not clear to us how to ablate this component immediately. Nevertheless, We are more than happy to hear further discussion from the reviewer.
>
> On the choice of the Gradient annealing hyper-parameters,  we treat it as a hyper-parameter that regularizes the strength of projection. In our experiments, we performed a hyper-parameter search for different annealing values and use the best one on validation data. Please refer to Appendix 7.4 and 7.5 to see the exact value used for each experiment.
>
> * **Regarding intuition on why FTP is better than TPGM.**
>
> FTP improves TPGM algorithmically in terms of efficiency, in terms of flexibility by removing limiting factors such as the need for validation data and nested update loops, and in some cases in terms of OOD robustness.  The differences in performance fluctuations can be partially attributed to the algorithmic differences between the two and also see discussion in Appendix 7.3, as well as simplified tuning, but we agree that future work could have the potential to understand and design even better methods.
>
> * **Regarding the continual learning experiments**
>
> Our comparison is absolutely fair because our method does not require knowledge of the task id during inference. It is privileged only to the knowledge of task boundary, which is also required in LwF.MC [45], L2P [46], DualPrompt [47], CODA-P [44], EWC [48], and L2 [44]. Specifically,
> * LwF changes the reference regularization model at the task boundary and treats new and old logit heads differently.
> * DualPrompt and Coda-p initialize new model parameters at the task boundary
> * L2P/DualPrompt/Coda-P treat old and new logits differently
> * EWC and L2 use task boundaries to determine the regularization parameters. EWC actually does significant regularization calculations at each task boundary.
>
> In other words, we do not use any additional information beyond what other compared methods do.

---

> > ### Comment · Reviewer_yjP3 · 2023-08-18
> >
> > The rebuttal address my concerns and I would like to maintain my rate.

---

### Official Review · Reviewer_wzxc · 2023-07-07

**Soundness:** 3 good
**Presentation:** 3 good
**Contribution:** 3 good
**Rating:** 5
**Confidence:** 2

**Summary:**

This paper introduces Fast Trainable Projection (FTP), a new algorithm aimed at improving the robustness and efficiency of fine-tuning pre-trained models. FTP achieves visible improvement in terms of computational speed and adaptability, demonstrated across various vision tasks and models, contributing to an average 35% speedup on benchmarks compared to prior works. The authors also provide a theoretical explanation for FTP's ability to maintain the robustness of pre-trained models through the lens of Lipschitz continuity.

**Strengths:**

1. papers are well-written, figures are well-made and very easy to read
2. problem being studied is important -- the efficient finetuning of models are particularly relevant given the popularity of foundation models today.

**Weaknesses:**

1. Limited to Vision Tasks: The experiments are largely focused on vision tasks. Additional experiments in other application areas such as natural language processing could further demonstrate the algorithm's effectiveness.
2. Does the method work for foundation-level models? For example, in vision, there's stable diffusion, segment anything etc. There are already very mature finetuning pipeline and benchmarks for these models that can be experimented with. If the proposed method works for these models, the impact will be significantly increased.

**Questions:**

1. Does the proposed algorithm work on low-rank adaptation (LoRA)? Would be interesting to see related experiments for that.

**Limitations:**

The authors have addressed the limitations

---

> ### Author Rebuttal · Authors · 2023-08-09
>
> We thank the reviewer for the positive and constructive feedback!
>
> * **Regarding limited-to-vision tasks**
>
> Theoretically, our method applies to any fine-tuning problems with pre-trained weights. To further showcase the adaptability of our method, we conducted a quick experiment on NLP for the rebuttal. Specifically, we use the DistilBert model [1], pre-trained on Wikipedia and Bookcorpus, and fine-tune it to 1% of the IMDB dataset. We finetune the DistilBert model in an unsupervised masked fashion and measure its performance on the fine-tuned IMDB dataset and the original Wikipedia dataset using perplexity (lower the better). We use the implementation open-sourced by Hugginface [2] for this experiment (15% token masking rate, 50 epochs, and learning rate 5-e5).
>
> |         | IMDB $\downarrow$ | Wikipedia  $\downarrow$ |
> |---------|-------|-----------|
> | Vanilla | 11.05 | 9.94      |
> | FTP     | **10.88** | **9.69**     |
>
> We use 1% data to simulate the situation of overfitting due to a small amount of data. Compared to vanilla fine-tuning, FTP can better avoid overfitting (lower perplexity on IMDB) and maintain more information from the original pre-training dataset (lower perplexity on Wikipedia). These observations corroborate our findings in other experiments.
>
> [1] Sanh, Victor, et al. "DistilBERT, a distilled version of BERT: smaller, faster, cheaper and lighter." arXiv preprint arXiv:1910.01108 (2019).
>
> [2] https://huggingface.co/learn/nlp-course/chapter7/3?fw=pt
>
> * **Regarding other foundation models**
>
> Theoretically yes. FTP is a generic projection-based optimization technique that can be integrated into existing optimizers to formulate new generic optimizers such as SGDP (sec.5). In this regard, theoretically, FTP can be applied to any fine-tuning problems (with pre-trained weights) in deep learning regardless of the underlying model architecture. In our experiments, we have successfully applied it to different deep learning models (CNNs and Transformers), tasks (classification and dense vision tasks), and even different learning paradigms (supervised, unsupervised and continual learning). While the rebuttal did not leave sufficient time to try additional foundation models, we agree this is a great area of future work to maximize impact, and we plan on doing so.
>
> * **Regarding LoRA.**
>
> No, the method does not apply to LoRA because LoRA adds additional weights that are not pre-trained and randomly initialized. Our method relies on the projection toward pre-trained weights. If a method introduces new model parameters, then our method does not apply.
>
> Nevertheless, we add experiments to compare to low-rank finetuning methods (as suggested by Reviewer 2kDG), Polyhistor [1], and LORA [2], using Adapter and low-rank factorization.  Specifically, we compare them on the PASCAL dense vision benchmarks in Sec.4.2.  In the following table, we directly compare them to the reported results from Polyhistr since we used its open-sourced code to benchmark FTP in our paper.
>
> |            | Segmentation $\uparrow$ | Human Parts $\uparrow$ | Surface Normal $\downarrow$|
> |------------|--------------|-------------|----------------|
> | Vanilla    | 66.03        | 62.21       | 18.98          |
> | LORA       | 70.12        | 57.73       | 18.96          |
> | Polyhistor | 70.87        | 59.54       | 17.47          |
> | FTP        | **73.79**        | **65.50**       | **15.51**          |
>
> We observe that on all fine-tuning tasks, FTP achieves better performance than LORA and Polyhistor.
>
> [1] Liu, Yen-Cheng, et al. "Polyhistor: Parameter-efficient multi-task adaptation for dense vision tasks." Advances in Neural Information Processing Systems 35 (2022): 36889-36901.
>
> [2] Hu, Edward J., et al. "Lora: Low-rank adaptation of large language models." arXiv preprint arXiv:2106.09685 (2021).

---

### Official Review · Reviewer_2kDG · 2023-07-08

**Soundness:** 2 fair
**Presentation:** 3 good
**Contribution:** 2 fair
**Rating:** 5
**Confidence:** 4

**Summary:**

In this paper, the authors propose an efficient fine-tuning method for deep learning models that learns the constraint for each layer more efficiently.
During neural network training, the proposed method optimizes the layer-wise constraint using the last batch of data, unlike a prior method (TPGM) that updates layer-wise constraints on a "validation" dataset. This new method is less computationally burdensome since the gradient of the constraint can be computed using the chain rule, which reuses the gradient stored for the last batch.
The authors also propose a notation to describe fine-tuning robustness and provide theoretical justification for the proposed method under this notation. To demonstrate the effectiveness of the new method, the authors conduct a comprehensive list of experiments and show that the proposed method is not only more efficient but also has better performance.

**Strengths:**

-The writing in this paper is of high quality, as the authors have presented the methodology in a clear manner.

-The related work section is well-discussed. The authors have categorized the existing fine-tuning studies into three categories: when, where, and how much to fine-tune. This is an insightful way to approach the current studies.

-The authors have provided theoretical justification for the proposed method.

-The experimental results are comprehensive, as the authors have included a list of different datasets and model architectures. The proposed method has been shown to outperform all baseline methods.

**Weaknesses:**

-My first concern is that the proposed method's novelty might be overshadowed due to its similarity to a previous method called TPGM. The major difference between the two seems to be the selection of batch data for updating the constraint parameter. The previous work used validation data, which may not be a common choice in the supervised learning setting, while this work uses the previous batch data. Although using the stored gradient indeed improves computational efficiency, the authors did not clearly state why this method has better performance than TPGM. The validation data and the other training data should have the same distribution.

-The proposed difference function's meaning requires in-depth justification. While (Lipschitz) smoothness is usually associated with better performance and (adversarial) robustness for deep neural networks, it is not directly implied that the proposed difference function will improve the downstream performance of a fine-tuned model. Additionally, this notation only describes the difference in feature space; how does it describe the linear-probing fine-tuning method, which only changes the linear last layer? Can it be shown that this proposed notation indicates fine-tuning downstream performance?

-The robustness notation here seems to be related to the OOD generalization capability. Are they the same thing here? Usually, the robustness is used to describe the model’s performance under some unwanted perturbation (adversarial samples or corruption)

-The Low-rank fine-tuning methods have been really popular recently. Is it possible to compare the proposed method with them?

Minor typo:

 Line 52, Liptschitz → Lipschitz

Line 127, “Then FTP calculates the gradients”, is here FTP meant to be TPGM?

**Questions:**

What’s the pattern of learned constraint for each layer, with regard to training epochs? Do they converge during the training?

I am really curious about the statement of discrepancies here. The author mentioned multiple times that the discrepancy between the validation data and training data or between the different batches of data enables the learning of meaningful projection constraints but without further elaborations. Typically, the training and validation data should be subject to the same distribution.

---

> ### Author Rebuttal · Authors · 2023-08-09
>
> We thank the reviewer for the constructive feedback!
>
> * **Regarding Novelty**
>
> The novelty of our method lies in its algorithmic improvement to the prior work TPGM in terms of efficiency, flexibility, and robustness. To successfully use the previous information to update internal parameters in FTP requires rigorous analysis. It requires careful caching of the correct model components from previous steps. For example, in Eq.4, we store previous “unconstrained” update $\tilde{W}_{t-1}$ instead of naively the previous update $W_{t-1}$ for mathematical correctness that’s not immediately clear without analysis. While the end result is a simple algorithm (a strength), our contribution is to come up with this method and rigorously evaluate it, showing significant benefits as other reviewers have mentioned.
>
> The differences in performance can be partially attributed to the algorithmic differences between the two and also see discussion in Appendix 7.3, but we agree that future work could have potential to understanding and designing even better methods.
> In addition to this, the improvement of FTP is also in its efficiency and flexibility.  For example, we demonstrated the regularization strength of FTP in continual learning experiments (sec.4.3). This is not possible with TPGM due to the lack of validation data under a continual learning setting. Hence such differences have practical import.
>
> * **Regarding the difference function**
>
>  Optimizing the difference function improves the robustness of the fine-tuned model. To understand the logic, we need to combine both Lemma 1 and Lemma 2.
>
> * Lemma 1 says that if we can minimize the right-hand side of the difference function, we can maximally maintain the robustness of the pre-trained model.
> * Lemma 2 says that minimizing the difference function, in fact, leads to a projection operation in the weight space. Therefore, the conclusion can be drawn that projection can lead to better fine-tuning robustness.
>
> The main contribution of the difference function is that by virtue of Lemma 2, it establishes an equivalence between feature space and weight space, i.e., projection in weight space is equivalent to minimizing the difference function in feature space. This applies to any weight layer including the last linear layer in a neural network.
>
> * **Regarding low-rank methods**.
>
> This is a great point, and for this rebuttal, we add experiments to compare to low-rank finetuning methods, Polyhistor [1] and LORA [2], using  Adapter and low-rank factorization.  Specifically, we compare them on the PASCAL dense vision benchmarks in Sec.4.2.  In the following table, we directly compare them to the reported results from Polyhistor since we used its open-sourced code to benchmark FTP in our paper.
>
> |            | Segmentation $\uparrow$ | Human Parts $\uparrow$ | Surface Normal $\downarrow$|
> |------------|--------------|-------------|----------------|
> | Vanilla    | 66.03        | 62.21       | 18.98          |
> | LORA       | 70.12        | 57.73       | 18.96          |
> | Polyhistor | 70.87        | 59.54       | 17.47          |
> | FTP        | **73.79**        | **65.50**       | **15.51**          |
>
> We observe that on all fine-tuning tasks, FTP achieves better performance than LORA and Polyhistor.
>
> [1] Liu, Yen-Cheng, et al. "Polyhistor: Parameter-efficient multi-task adaptation for dense vision tasks." Neurips (2022)
>
> [2] Hu, Edward J., et al. "Lora: Low-rank adaptation of large language models."  (2021).
>
> * **Regarding FTP learned pattern**
>
> Please see the attached PDF for a visualization.
> The learned constraints grow slowly from a small value, e.g., 1e-8, at different paces for each layer and eventually converge. The pattern is highly similar to that of TPGM as reported in its paper [1]. We observed that layers closer to the output tend to have larger constraints (less constrained and more changes allowed) whereas the layers closer to the input tend to have smaller constraints (more constrained). This is similar to our intuition that early layers learn more general information and later layers learn task-specific information. The constraints do converge because they are calculated based on the current model gradient g_t which decreases to zero as the learning rate drops (Eq.4).
> To qualitatively show this, we record the history of FTP constraints for the classification experiment in Appendix Tab.6. In this setting, we fine-tune a pre-trained ResNet50 on DomainNet real images. We visualize the learned constraints for each layer through time in the attached pdf file. There are two observations. 1) Early layers (dark colors) generally have smaller constraints than the latter layers (light colors) throughout training. 2) Constraints grow from small to large and converge in the end.
>
> [1] Tian, Junjiao, et al. "Trainable Projected Gradient Method for Robust Fine-tuning." CVPR 2023.
>
> * **Regarding "discrepancies"**
>
> This is a great question. Here the discrepancies refer to the variances between sampled batches. Even though training data and validation data are sampled from the same distribution, each mini-batch can lead to very different directions of updates on the current model. Intuitively, we utilize these variances to “control” how much the model should deviate from the pre-trained weights. If two batches disagree more on where the gradients should go in a certain direction, then the projection will be stronger towards the pre-trained model.
> Mathematically, this is reflected by the dot product in the updated equation 4 between
> $\mathbf{g}_t^{i,\intercal}$
>
> and $-({\tilde{\mathbf{w}}_{t-1}^i}-\mathbf{w}_0^i)$ where the first quantity represents the gradients of the current batch and the second quantity represents the gradients of the previous batch. If the two batches disagree, then the product will be positive, which leads to smaller projection constraints (i.e., the gradient $\nabla\gamma_t$ is positive), i.e., stronger projection.

---

> > ### Comment · Reviewer_2kDG · 2023-08-20
> > **Response**
> >
> > I would like to thank the authors' response as well as the additional experimental results. While I still hold my concern regarding some of the limitations I raised here, (most importantly, the discrepancies between training/validation batches), most of my concerns have already been addressed and I will increase my score and lean to acceptance. Thanks!

---

### Official Review · Reviewer_HDd6 · 2023-07-12

**Soundness:** 3 good
**Presentation:** 3 good
**Contribution:** 2 fair
**Rating:** 5
**Confidence:** 5

**Summary:**

The paper presents a new algorithm for robust fine-tuning of pre-trained models, specifically focusing on maintaining out-of-distribution (OOD) robustness while achieving competitive in-distribution (ID) performance. The algorithm, dubbed Fast Trainable Projection (FTP), is designed to overcome the scalability and efficiency limitations of current methods, offering a 35% speedup compared to prior work (to be specific, TPGM). FTP achieves this by efficiently learning per-layer projection constraints during the fine-tuning process.

The claimed contributions of this paper are:

1. The introduction of the FTP algorithm that significantly improves computational efficiency while learning projection constraints and fine-tuning the model simultaneously. This algorithm can be integrated with existing optimizers like SGD and AdamW and can be adopted as a new drop-in fine-tuning optimizer.

2. Empirical validation of the FTP algorithm's robustness on OOD datasets. They tested the algorithm across four vision tasks with five different pre-trained models, demonstrating robustness especially in scenarios with domain shifts and natural corruptions. The FTP algorithm also achieves state-of-the-art performance on a continual learning benchmark.

3. Theoretical explanation of FTP's robustness maintenance capabilities. The authors provide a mathematical perspective that explains why FTP is effective at preserving the robustness of pre-trained models. They explore this through the lens of Lipschitz continuity, taking into account both the feature space and weight space of a model.


**Strengths:**

The introduction of the Fast Trainable Projection (FTP) algorithm is an interesting contribution. It addresses the real-world limitations of scalability and efficiency associated with TPGM, making it a practical solution for various tasks.

The robustness of the FTP algorithm has been validated with experiments across four vision tasks and five pre-trained models. Superior results on OOD datasets and state-of-the-art performance on a continual learning benchmark provide empirical support for the authors' claims.

**Weaknesses:**

1. **Limited Scope:** the proposed method focuses on a specific Trainable projected gradient method (TPGM) and aims to improve its efficiency. However, the extent of its practical application could be questioned as it's not entirely clear how well it would generalize to other types of models or algorithms. Its significance might be seen as limited if it only optimizes a specific kind of model, especially when there are many alternative and possibly more efficient methods available.

2. **Lack of Efficiency:** Despite the fact that the FTP improves on the TPGM's efficiency, it remains slower than many other methods, which raises questions about its practicality in real-world applications where computational resources and processing times are often crucial factors.

3. **Theoretical Contributions:** The theoretical contribution of this paper may be seen as somewhat trivial, as the authors basically just formalize an intuitive understanding of Lipschitz continuity in relation to model robustness. The assumption that robustness equates to Lipschitz continuity may oversimplify the complex nature of robustness in deep learning models. Real-world robustness often depends on various other factors such as data variance, architecture, and loss landscape, which aren't addressed in this paper. Furthermore, they fail to delve deeper into the specifics of how the generalization relates to pre-trained weights, which might have offered more novel insights.

**Questions:**

What is the memory footprint and runtime overhead of the proposed method?

How does the proposed method affect generalization of neural networks?

**Limitations:**

Yes.

---

> ### Author Rebuttal · Authors · 2023-08-09
>
> We thank the reviewer for the constructive feedback!
>
> * **Regarding limited scope**
>
> Flexibility and extended applicability are advantages of the proposed method FTP because FTP is a model-agnostic optimizer.  Specifically, FTP is a generic projection-based optimization technique that can be integrated into existing optimizers such as Adam and SGD to formulate new generic optimizers such as AdamP and SGDP (Sec.5 and Appendix 7.7). In this regard, theoretically, FTP can be applied to any fine-tuning problems in deep learning regardless of the underlying model architecture. In our experiments, we have successfully applied it to the below models. **Note that we have added an entirely new task/model result for masked language learning, demonstrating the significant flexibility of our method.**
> * Different deep-learning models
>   * CNNs (Tab.1, 2, 6 )
>   * Transformers (Tab.3, 4, 5,  7, 8)
> * Different Tasks
>   * Classification (Tab.1, 2, 6)
>   * Dense classification
>     * Segmentation (Tab. 4)
>     * human parts segmentation (Tab.7)
>   * Dense regression
>     * Surface normal estimation (Tab.8)
>    * Masked language learning
>      * See discussion with Reviewer WZxc.
> * Different learning paradigms
>   * Supervised learning(Tab. 1, 2, 3, 4, 6, 7, 8)
>   * Continual learning (Tab.5)
>   * Unsupervised learning(See discussion with Reviewer WZxc)
>
> [1] Sanh, Victor, et al. "DistilBERT, a distilled version of BERT: smaller, faster, cheaper and lighter." (2019).
>
> [2] https://huggingface.co/learn/nlp-course/chapter7/3?fw=pt
>
> * **Regarding efficiency**
>
> As mentioned by Reviewer Fz4t, the benefits of FTP are many-fold including better OOD generalization and lack of need of held-out validation data. Further, in terms of efficiency as mentioned computation tends to be smaller than the large-scale pre-training, so some slowdown can be tolerated. More specifically, FTP’s efficiency can be optimized to suit different practical fine-tuning applications that may vary what is finetuned. To see this, we need to understand that FTP is an optimizer that can be applied to any gradient descent-based fine-tuning problems. Just like we can fine-tune an entire model using SGD or just fine-tune the bias terms for faster speed (Partial Fusion in Tab.1 and Tab.2), FTP can be used to just fine-tune the bias terms as well, which will lead to much better efficiency. For example, in the continual learning experiments (Sec.4.3), FTP is used only to fine-tune the QKV components in a Transformer model. To further demonstrate the point, for the rebuttal here we show the efficiency of applying FTP to different parts of a model.  Specifically, we adopt the same setting as in the segmentation experiment (Tab.4), which uses a Swin-Transfomer as the backbone.  Here we only profile the time used by the optimization process, excluding data loading and the model forward pass, which are not part of FTP,  to more directly demonstrate the efficiency of FTP.
>
> | s/it | Full Model | Bias Only | QKV Only | Decoder Only |
> |------|------------|-----------|----------|--------------|
> | Adam | 0.204      | 0.187     | 0.169    | 0.168        |
> | FTP  | 0.314      | 0.280     | 0.188    | 0.186        |
>
> There are two important observations. 1) on average FTP is only 30% slower than the vanilla optimizer, which is not a major bottleneck in most cases. 2) as the number of tuning parameters decreases, the speed difference between FTP and the vanilla optimizer further diminishes (only 10% slower when only tuning the decoder). **This means that in extreme cases where computation resources and processing times are critical, FTP is virtually just as fast as vanilla optimizers for example when only finetuning a single last linear layer as in linear probing.**
>
> * **Regarding Theory**
>
>  Lipschitz continuity is a standard and popular mathematical measurement of robustness in deep learning literature [1,2]. While it does not encompass all the aspects of deep learning robustness such as data variance, architectures, etc., it is a valuable mathematical tool to start analyzing the robustness of deep learning models. The generalization capability of a pre-trained model in its relationship to the pre-trained weights is explored in the prior work [3] through linear systems.  Further, we believe our extremely thorough empirical results of a practical algorithm bolster the practicality of the analysis, despite limitations.  In short, the better the pre-training datasets cover the downstream datasets, the more robust the fine-tuned model will be.
>
> [1] Weng, Tsui-Wei, et al. "Evaluating the robustness of neural networks: An extreme value theory approach." (2018).
>
> [2] Zhang, Bohang, et al. "Rethinking Lipschitz neural networks and certified robustness: A boolean function perspective."Neurips (2022): 19398-19413.
>
> [3] Tian, Junjiao, et al. "Trainable Projected Gradient Method for Robust Fine-tuning." CVPR. 2023.
>
> * **Regarding memory**
>
> The only major memory requirement is caching the previous gradient. This is rather a common requirement. For example, the Adam optimizer keeps running copies of aggregated first and second-moment information from previous iterations.
> The memory consumption consumed by storing the projection constraints is negligible since they are scalars, the size of which are orders of magnitude smaller than the size of gradients.
>
> * **Regarding Generalization**
>
> FTP has a positive effect on the generalization of neural networks as seen in our OOD generalization experiments in Tab.1 and Tab.2, where FTP achieves the best performance. The goal of the proposed optimizer is to improve the generalization ability of the fine-tuned model by maintaining the knowledge acquired during pre-training. In this regard, the effects on generalization should scale positively with the generalization ability of the pre-trained model. In other words, if we use a stronger pre-trained model, the positive effect on generalization from FTP will be even bigger.

---

> > ### Comment · Reviewer_HDd6 · 2023-08-15
> >
> > I have raised my score, to appreciate the empirical results of this paper.
> >
> > The fact that it is just an addon of a previous paper that is not widely used, remains unchanged.
> >
> > And I insist that the "theoretical" analysis part should be removed if the paper is finally accepted. It indeed adds little value to the paper.

---

### Official Review · Reviewer_FZ4t · 2023-07-27

**Soundness:** 3 good
**Presentation:** 3 good
**Contribution:** 2 fair
**Rating:** 6
**Confidence:** 4

**Summary:**

This paper presents Fast Trainable Projection (FTP), an efficient fine-tuning algorithm. FTP learns a projection radius based on the current training batch. Through both analysis and experiments, the paper shows that FTP improves out-of-distribution robustness while maintaining in-distribution performance. Experiments also show that FTP can accelerate learning by up to 35% compared to previous methods.

**Strengths:**

- The idea seems to be a natural improvement following the lines of MARS-SP (project after unconstrained GD step) and TPGM (learn a different radius for each layer).
- Section 3.3 was clear and interesting; it establishes a connection between weight-space projection and "robustness" in terms of the Lipschitz constant.
- The experimental results are strong, across DomainNet, ImageNet, and PASCAL fine-tuning experiments.

**Weaknesses:**

- A main claimed benefit of FTP (in the abstract and intro) is its computational efficiency. To what extent is computational cost a bottleneck in the experimental settings you consider? I think fine-tuning is generally considered to be pretty computationally light, especially compared to the training of the foundation model itself. Given the experimental results, maybe the more relevant benefits are (1) better OOD acc (2) no need for held-out val data like TPGM.
- While section 3.3 was interesting, I'm not sure if it contributes the to the main point of the paper (benefits of FTP over e.g TPGM), since this simplified analysis really applies to all projection-based methods. Does this analysis motivate FTP over other ways of using projection for fine-tuning?
minor typo: Fig 2 caption PorjUpdate -> ProjUpdate

**Questions:**

- There seem to be similarities between FTP and the general hyperparameter tuning method proposed in https://arxiv.org/abs/1909.13371, in that both methods delay updates with the current batch to optimize hyperparameters in an online fashion (gamma here corresponds to alpha there). Could the authors comment on how these two methods relate?
- see other questions in "Weaknesses" above

**Limitations:**

Yes.
No particular negative societal impact.

---

> ### Author Rebuttal · Authors · 2023-08-09
>
> We thank the reviewer for the positive and constructive feedback!! We have added additional results/visualization according to other reviewers' comments. We hope they would further strengthen your confidence in our method.
>
> * **Regarding Benefits of FTP**
>
> Indeed, fine-tuning is much less computationally heavy than pre-training. In our experiments, fine-tuning on ImageNet takes about 2 days using TPGM but roughly 1 day using FTP. The speed-up of FTP vs. TPGM is roughly 2x. Nevertheless, as the reviewer mentioned, FTP brings other benefits such as no need for held-out validation data. This is a major improvement because it allows us to integrate FTP into existing optimizers for much better adaptability. For example, we demonstrated the regularization strength of FTP on continual learning experiments (sec.4.3). This is not possible with TPGM due to the lack of validation data under a continual learning setting. Thank you for the suggestions; we will better emphasize all of the benefits in the revised version!
>
> * **Regarding Sec 3.3**
>
> Thank you for pointing out the typo. No, the theory does not indicate that FTP is better than TPGM. The improvement of FTP over TPGM is algorithmic in terms of efficiency and applicability, not theoretical.
> Nevertheless, section 3.3 introduces a general theory on why projection is useful for fine-tuning, a theoretical question that hasn’t been answered in prior works. While you are right that it generally applies to all projection-based methods including FTP and TPGM, we believe it was important to include to progress this area empirically and theoretically. We will add text and organization to explicitly discuss this.
>
> * **Regarding https://arxiv.org/abs/1909.13371**
>
> Thank you for bringing up this work, which is a very valuable reference for FTP. We will cite and discuss this work. The work introduces a smart backpropagation modification for computing “hyper gradients” to optimize internal hyper-parameters in an existing optimizer. From this perspective, FTP can be seen as an extension of this idea of projection-based optimization, where the projection constraints are the hyper-parameters.
> Nevertheless, our novelty stands out in two aspects. First, FTP introduces the projection operation as an integral component of an optimizer. The projection has always been treated as a separate operation outside the update of optimizers [1,2]. It is not clear how to apply the idea of “hyper-gradient” unless we formalize projection as part of the gradient update as in this paper. Second, the update of “gamma” uses the Adam update rule and is not simply gradient descent as in the referenced work. The Adam update rule introduces smoothness in the updates of the projection parameters and enables us to apply the updated projection parameters to the current updated model without rolling back to the previous state, greatly saving computation (please see Appendix 7.3 for detailed discussion).
>
>
> [1] Gouk, Henry, Timothy M. Hospedales, and Massimiliano Pontil. "Distance-based regularisation of deep networks for fine-tuning." arXiv preprint arXiv:2002.08253 (2020).
>
> [2] Tian, Junjiao, et al. "Trainable Projected Gradient Method for Robust Fine-tuning." Proceedings of the IEEE/CVF Conference on Computer Vision and Pattern Recognition. 2023.

---

### Author Rebuttal · Authors · 2023-08-09

We thank the reviewers for all the constructive feedback and positive comments on the real-world relevance (wzxc), strong performance (Fz4t), and extensive experiments (HDd6, 2kDG, yjp3). We have provided a detailed discussion for each of your questions below.
For this rebuttal, we clarified some misunderstandings and introduced new experiments to address specific questions.

To recap, we proposed FTP to improve the generalization and robustness of fine-tuned models. FTP is
* An optimization technique that can be integrated into existing optimizers.
* **Easy to use** in a plug-and-play fashion.
* **Broadly applicable** to most fine-tuning problems regardless of the underlying tasks and models.
* **More efficient** than prior works, achieving ~2x speedup.

We have demonstrated its effectiveness across different deep learning models (CNNs and Transformers), tasks (classification and dense vision tasks), and even different learning paradigms (supervised, unsupervised and continual learning).



New experiments:

* Masked Langage modeling experiments (suggested by Reviewer WZxc)
* The efficiency of FTP when fine-tuning to different parts of a model (inspired by Reviewer HDd6)
* Comparison to low-rank fine-tuning methods e.g., LoRA (suggested by Reviewer 2kDG)
* Visualization of FTP’s learned per-layer constraints with respect to training time (suggested by Reviewer 2kDG)

All these experiments are included in the attached PDF file.

---

### Decision · Program_Chairs · 2023-09-21

**Decision:**

Accept (poster)

**Comment:**

All of the reviewers are positive about the paper, and the rebuttal helped clarify key points. The reviewers noted the strong empirical results.